# Quantum simulation of the bosonic Kitaev chain

**Jamal H. Busnaina[1], Zheng Shi [1], Alexander McDonald[2,3], Dmytro Dubyna [1], Ibrahim Nsanzineza[1], Jimmy S. C. Hung [1], C. W. Sandbo Chang[1], Aashish A. Clerk [2] & Christopher M. Wilson [1]** ✉

Superconducting quantum circuits are a natural platform for quantum simulations of a wide variety of important lattice models describing topological phenomena, spanning condensed matter and high-energy physics. One such model is the bosonic analog of the well-known fermionic Kitaev chain, a 1D tight-binding model with both nearest-neighbor hopping and pairing terms. Despite being fully Hermitian, the bosonic Kitaev chain exhibits a number of striking features associated with non-Hermitian systems, including chiral transport and a dramatic sensitivity to boundary conditions known as the non-Hermitian skin effect. Here, using a multimode superconducting parametric cavity, we implement the bosonic Kitaev chain in synthetic dimensions. The lattice sites are mapped to frequency modes of the cavity, and the in situ tunable complex hopping and pairing terms are created by parametric pumping at the mode-difference and mode-sum frequencies, respectively. We experimentally demonstrate important precursors of nontrivial topology and the non-Hermitian skin effect in the bosonic Kitaev chain, including chiral transport, quadrature wavefunction localization, and sensitivity to boundary conditions. Our experiment is an important first step towards exploring genuine many-body non-Hermitian quantum dynamics.

While the development of universal fault-tolerant quantum computers is still ongoing, analog quantum simulation (AQS) has emerged as a promising approach to study classically intractable quantum systems[1–3]. In AQS, the simulations take place on an artificial quantum system built to have the same Hamiltonian as the system of interest. One appealing aspect of AQS is that quantum degrees of freedom can be represented natively. For instance, while bosonic degrees of freedom with infinite Hilbert spaces can be naturally simulated using oscillator modes on an AQS platform, boson-to-qubit mapping on a small-scale qubit-based computer typically requires a very large overhead in the number of physical qubits and required gates to implement bosonic operators[4–7].

An important class of AQS is lattice models with topological properties, which are intrinsically plagued by the infamous "sign problem" and thus are unsuitable for quantum Monte Carlo methods[8–10]. Topological systems have been a focus of intense research, both theoretically and experimentally, for some time. More recently the study of classical and quantum topological physics in non-Hermitian systems has attracted significant interest[11,12]. These systems exhibit intriguing and distinct properties owing to their nonorthogonal eigenstates and singularities in the complex eigenvalue spectrum of non-Hermitian matrices[13]. Compelling phenomena include oscillations between eigenstates[14,15], unidirectional invisibility[16], high-performance lasers[17], and enhanced sensitivity for potential sensing applications[18].

[1]Institute for Quantum Computing and Department of Electrical & Computer Engineering, University of Waterloo, Waterloo, ON N2L 3G1, Canada. [2]Pritzker School of Molecular Engineering, University of Chicago, Chicago, IL 60637, USA. [3]Institut quantique and Département de Physique, Université de Sherbrooke, Sherbrooke, QC J1K 2R1, Canada. ✉e-mail: chris.wilson@uwaterloo.ca

Remarkably, non-Hermiticity fundamentally alters concepts such as symmetry and energy gaps inherited from Hermitian physics, giving rise to an enriched variety of topological phases with no Hermitian counterpart[13].

There are various approaches to non-Hermitian dynamics[19–22]. In the quantum regime, non-Hermiticity appears naturally in open quantum systems described by a Lindblad master equation: conditioned on the absence of a quantum jump, state evolution is described by an effective non-Hermitian Hamiltonian[23,24]. Experimentally resolving potentially interesting effects from such non-Hermitian Hamiltonians, while possible[25–27], is challenging. By definition, one must post-select on measurement records without a jump, and such trajectories become exponentially rare at long times. Consequently, one must perform many runs of the experiment to acquire enough data for adequate statistics[28]. An alternate route to obtaining non-Hermitian quantum dynamics is through the use of Hermitian bosonic Hamiltonians. With unitary squeezing and antisqueezing terms, the equations of motion become effectively non-Hermitian despite the Hermiticity of the underlying Hamiltonians[29–31]. These Hamiltonians present an interesting avenue for probing coherent, genuinely quantum non-Hermitian effects, without the use of dissipation[32].

Previously, we have demonstrated the feasibility of an AQS platform based on a multimode superconducting parametric cavity by simulating a plaquette of the bosonic Creutz ladder[33]. The cavity modes share a superconducting quantum interference device (SQUID) which acts as a common boundary condition. Parametric modulation of the boundary condition induces complex "hopping" couplings that allow us to create a programmable graph of connected (coupled) modes, realizing a lattice in synthetic dimensions. By controlling the phases of the complex hopping terms, we showed that our platform can implement interesting features including static gauge fields and topological phenomena. We note similar work has been done in the context of classical non-Hermitian optics[34,35] and chiral photon transport[36,37].

In this work, we expand our programmable AQS toolbox by introducing pairing terms between modes in the target Hamiltonian in addition to the hopping terms. They allow us to construct a topologically nontrivial Hamiltonian in synthetic dimensions, the bosonic Kitaev chain (BKC) introduced in ref. 38. Working with a 3-site chain, we experimentally demonstrate phase-dependent chiral transport, quadrature wavefunction localization, and a strong sensitivity to boundary conditions. These observations serve as precursors to nontrivial topology and the much sought-after non-Hermitian skin effect (NHSE)[32,39–41]. Our work enlarges the set of topologically nontrivial models that can be simulated under genuinely quantum conditions, further highlighting the potential of our AQS platform.

## Results

We now review the features of the BKC[38]. In analogy with the celebrated fermionic Kitaev chain[42], a spinless p-wave topological superconductor in one dimension, the BKC is a 1D bosonic tight-binding model with both nearest-neighbor hopping and pairing terms[38]. As a consequence of the pairing terms, the system has effectively non-Hermitian equations of motion and supports phase-dependent chiral transport. Furthermore, it has a topologically nontrivial phase where each band of the bulk spectrum of the dynamical matrix has nonzero winding in the complex energy plane. This is accompanied by the remarkable property of the NHSE[43–46]: the entire spectrum depends sensitively on boundary conditions, and the wavefunctions under open boundary conditions are localized at the ends of the chain. (See Supplementary Note 3 for a detailed discussion of the symmetry of the model,

the topological invariant, and the topological protection against symmetry-preserving disorder).

The BKC is described by the 1D tight-binding Hamiltonian

$$\hat{\mathcal{H}}_B = \frac{1}{2}\sum_j (te^{i\varphi_t}\hat{a}^\dagger_{j+1}\hat{a}_j + i\Delta\hat{a}^\dagger_{j+1}\hat{a}^\dagger_j + \text{h.c.}), \qquad (1)$$

illustrated in Fig. 1a. Here $\hat{a}_j$ is a bosonic annihilation operator on site $j$, and $te^{i\varphi_t}$ and $i\Delta$ are respectively the complex hopping and pairing strengths between adjacent sites, with magnitudes $t$ and $\Delta$. Without loss of generality, we choose the pairing strength to be purely imaginary.

To demonstrate that the model supports phase-dependent chiral transport, we consider the Heisenberg equations of motion for the Hermitian position and momentum quadratures $\hat{x}_j = (\hat{a}_j + \hat{a}^\dagger_j)/\sqrt{2}$ and

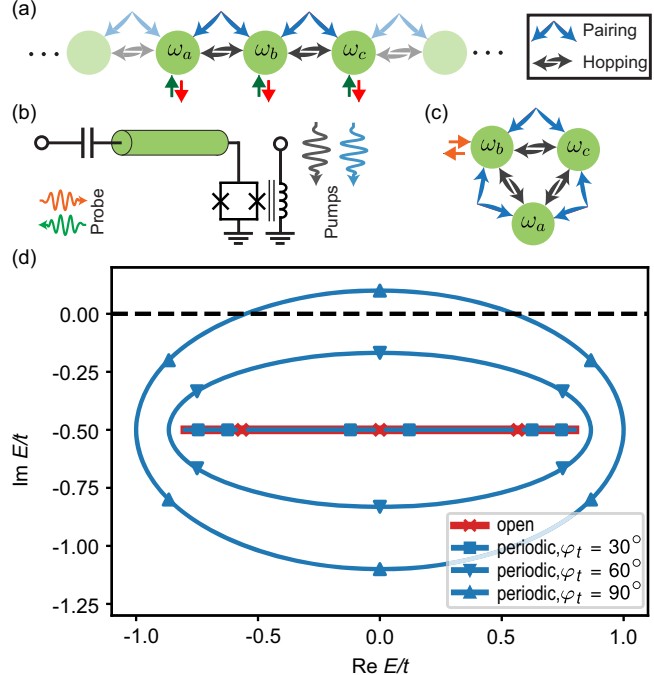

**Fig. 1 | Quantum simulation of the bosonic Kitaev chain. a** Schematic representation of the bosonic Kitaev chain (BKC). Black and blue arrows indicate the hopping and pairing couplings, respectively. **b** Device cartoon. The short circuit at one end of the resonator is replaced by a SQUID that creates a tunable boundary condition for the resonator modes. The cavity's fundamental mode is around 400 MHz, with 13 higher modes within the measurement bandwidth of 4–12 GHz. We achieve uneven spacing between cavity modes through impedance engineering[59,60]. This allows us to selectively activate couplings between desired pairs of modes. We create parametric interactions by pumping the SQUID through an on-chip flux line. The blue and black signals represent the two pumps that create complex hopping and pairing couplings, respectively. **a, c** Synthetic Kitaev lattices. We program 3-site BKCs in synthetic dimensions with (**a**) open and (**c**) periodic boundary conditions, using four and six pump tones, respectively. We probe the open chain (**a**) by sending a coherent tone at the frequency of one site and measuring the reflected and transported signals at all sites. The injected signal propagates through the chain and eventually leaks out, where it is then detected at all site frequencies via three RF digitizers. In the closed chain (**c**), we focus on the spectrum by measuring the reflection coefficient around the frequency of each site. **d** Complex spectrum of the $N$-site BKC with $\Delta/t = 0.6$ and $\kappa/t = 1$ under open and periodic boundary conditions, as predicted by Eqs. (4) and (6). Lines correspond to long chains with $N \gg 1$ and markers indicate the case $N = 3$. For periodic boundary conditions, each band of the spectrum has nonzero topological winding when $|\cos\varphi_t| < \Delta/t$. Further decreasing $|\cos\varphi_t|$ eventually drives the system into instability once the spectrum touches the real axis (dashed line).

$\hat{p}_j = -i(\hat{a}_j - \hat{a}_j^\dagger)/\sqrt{2}$:

$$\dot{\hat{x}}_j = \frac{1}{2\hbar}\left[(t\sin\varphi_t + \Delta)\hat{x}_{j-1} - (t\sin\varphi_t - \Delta)\hat{x}_{j+1} + t\cos\varphi_t\left(\hat{p}_{j-1} + \hat{p}_{j+1}\right)\right], \tag{2}$$

$$\dot{\hat{p}}_j = \frac{1}{2\hbar}\left[(t\sin\varphi_t - \Delta)\hat{p}_{j-1} - (t\sin\varphi_t + \Delta)\hat{p}_{j+1} - t\cos\varphi_t\left(\hat{x}_{j-1} + \hat{x}_{j+1}\right)\right]. \tag{3}$$

In the special case $\varphi_t = \pm 90°$, as long as $\Delta \neq 0$, the equations of motion correspond exactly to two decoupled non-Hermitian Hatano–Nelson chains with asymmetric left-right hopping for the $x$ and $p$ quadratures[47]. This is an example of effective non-Hermitian dynamics in Hermitian systems[31,48]. The decoupling of the $x$ and $p$ quadratures, along with the left-right coupling asymmetry in each chain, gives rise to phase-dependent chiral propagation, with quadratures representing the two chiral species. The chirality becomes perfect in the $\Delta \to t$ limit: for $\varphi_t = 90°$ ($-90°$), the $x$ ($p$) quadrature propagates to the right and the $p$ ($x$) quadrature to the left. From Eqs. (2) to (3), we see that away from $\varphi_t = \pm 90°$ a finite $\cos\varphi_t$ term mixes the left and right-moving quadratures, leading to an overall reduction in the chiral nature of the transport[38].

Let us consider the spectrum of the dynamical matrix, i.e., the coefficient matrix of Eqs. (2)–(3), taking into account a uniform onsite single-photon loss rate $\kappa$. Eqs. (2)–(3) are now understood as the Heisenberg-Langevin equations of the expectation values $\langle\hat{x}_j\rangle$ and $\langle\hat{p}_j\rangle$, and acquire the damping terms $-\kappa\langle\hat{x}_j\rangle/2$ and $-\kappa\langle\hat{p}_j\rangle/2$.

We can perform a spatially uniform squeezing transformation to identify a topological phase transition at $t|\cos\varphi_t| = \Delta$. For $t|\cos\varphi_t| > \Delta$ the transformed Hamiltonian for the squeezed fields $\hat{\beta}_j$ is equivalent to a model with only hopping. In the more interesting case of $t|\cos\varphi_t| < \Delta$, the transformed Hamiltonian assumes the form of Eq. (1) with renormalized parameters $t' = t\sin\varphi_t$, $\Delta' = \sqrt{\Delta^2 - t^2\cos^2\varphi_t}$ and $\varphi_t' = 90°$.

Exploring the second case further, we consider both open and periodic boundary conditions. The complex energy spectrum for the open chain reads

$$E_n^o = \sqrt{t^2 - \Delta^2}\cos k_n - i\frac{\kappa}{2}, \tag{4}$$

where $k_n = n\pi/(N+1)$, $n = 1, 2, \ldots, N$ and $N$ is the number of sites. The open chain spectrum Eq. (4) is independent of the coupling phase $\varphi_t$, and can be obtained through a second, position-dependent squeezing transformation[38]. This position dependence is reflected in the quadrature wavefunctions which do depend on $\varphi_t$:

$$\hat{d}_n \propto \sum_j \sin(k_n j)(e^{rj}\hat{x}_j' + ie^{-rj}\hat{p}_j'), e^{-2r} = \frac{|t' - \Delta'|}{t' + \Delta'}, \tag{5}$$

where $\hat{d}_n$ is the annihilation operator for the eigenstate corresponding to momentum $k_n$, $\hat{x}_j' = (\hat{\beta}_j + \hat{\beta}_j^\dagger)/\sqrt{2}$, and $\hat{p}_j' = -i(\hat{\beta}_j - \hat{\beta}_j^\dagger)/\sqrt{2}$. It is clear from the exponential factors that different quadrature components of a given wavefunction are localized at opposite edges of the chain in the topological phase.

The spectrum for periodic boundary conditions is found by Fourier transforming the dynamical matrix,

$$E_n^p = t\sin\varphi_t \sin k_n \pm i\sqrt{\Delta^2 - t^2\cos^2\varphi_t}\cos k_n - i\frac{\kappa}{2}, \tag{6}$$

where $k_n = 2n\pi/N$, $n = 1, 2, \ldots, N$. In the topological phase $t|\cos\varphi_t| < \Delta$, Eq. (6) forms an ellipse in the complex energy plane, yielding a nonzero winding number around the point $E = -i\kappa/2$ for each band. Comparing with Eq. (4), we see that the spectrum of the system depends drastically on its boundary conditions, a feature which holds for arbitrary system size (see Fig. 1d). Together with the (phase-dependent) localization in the open-chain wavefunctions in Eq. (5), it thus serves as a paradigmatic example of the NHSE[43,44].

Equations (4) and (6) imply that, provided

$$\sqrt{t^2\cos^2\varphi_t + \frac{\kappa^2}{4}} < \Delta < \sqrt{t^2 + \frac{\kappa^2}{4\cos^2\pi/(N+1)}}, \tag{7}$$

the system is dynamically stable under open boundary conditions (i.e., all energy eigenvalues have nonpositive imaginary parts), but unstable under periodic boundary conditions. Such instability in the presence of loss stems from the effective non-Hermiticity induced by pairing, and provides direct evidence for the nontrivial topology of the system.

We now describe the experimental realization of the BKC on our AQS platform (see Fig. 1b and ref. 33). We program a chain of three sites in the synthetic frequency dimension. For the open chain, the sites are connected by two links where each link is created by two coherent pumps: a pump at the modes' frequency difference $\omega_{j,j+1}^t = |\omega_j - \omega_{j+1}|$ to activate the hopping and a pump at the sum frequency $\omega_{jj+1}^\Delta = \omega_j + \omega_{j+1}$ to activate the pairing. The magnitudes and phases of these pump tones in turn determine the magnitudes and phases of the complex hopping and pairing terms. To impose periodic boundary conditions on the 3-site chain, we create an additional link with two more pump tones that connect the open ends, forming a closed chain (Fig. 1c).

We measure the spectra of both chains using a vector network analyzer (VNA), determining the eigenmode frequencies directly from the reflection coefficients. We further characterize the open chain using phase-sensitive transport measurements. Sending in a tone set at a constant magnitude but with a phase that ramps at a constant rate from $-180°$ to $180°$, we probe at various site frequencies to measure signal transport in synthetic dimensions. The phase-sensitivity of the transport converts the phase sweep of the input signal into magnitude variations in the output signals. Additional details on device fabrication and the measurement setup are given in Supplementary Notes 1 and 2.

## Twisted-tubes picture

Before discussing the details of our transport measurements, we present a twisted-tubes picture (Fig. 2) explaining the role of individual hopping and pairing phases. Experimentally, while all pump phases are separately tunable, they are difficult to calibrate absolutely at the sample, which is deep in the cryostat. The twisted-tubes picture is an intuitive way to understand how adjusting the pump phases affects the dynamics of the system.

We begin from a single link $a$–$b$ with hopping $t_{ab}e^{i\varphi_{ab}^t}$ and pairing $\Delta_{ab}e^{i\varphi_{ab}^\Delta}$. For future convenience, we define the sum and difference phases, $\varphi_{ab}^\pm = (\varphi_{ab}^t \pm \varphi_{ab}^\Delta)/2$. The transport properties are solved using a phase-dependent input-output theory (see Methods). The transported signal strength from $a$ to $b$ depends on the input phase. We refer to the input quadrature at $a$ that maximizes transport, together with the corresponding output quadrature at $b$, as the favored quadratures in transport from $a$ to $b$. On the other hand, the orthogonal quadratures dominate the transported signal in the opposite direction. This is represented in Fig. 2 as a pair of interleaved directional tubes with (squeezed) elliptic cross sections, where red (blue) tubes transport to the right (left). The favored quadratures in a given transport direction are along the major axis of the corresponding tube. We can twist the tubes by varying $\varphi_{ab}^-$ and $\varphi_{ab}^+$, which individually rotate the $a$ and $b$ ends, respectively (Fig. 2a).

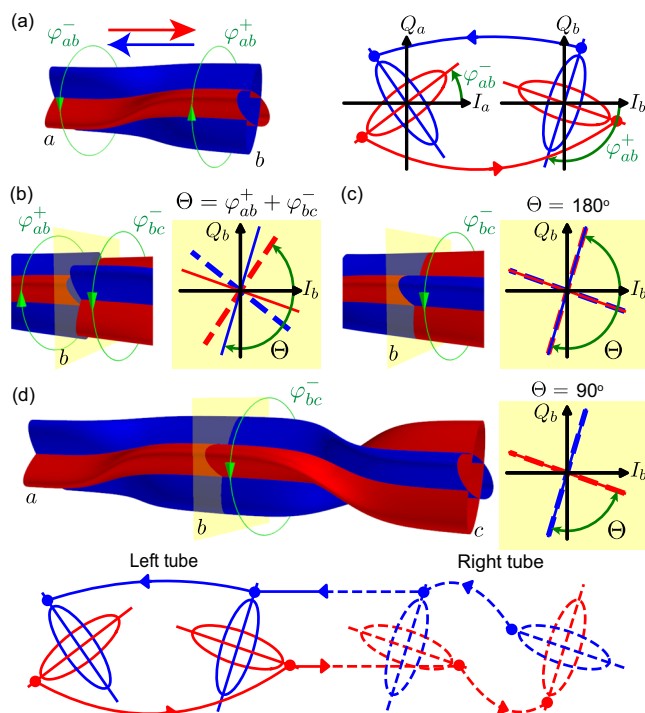

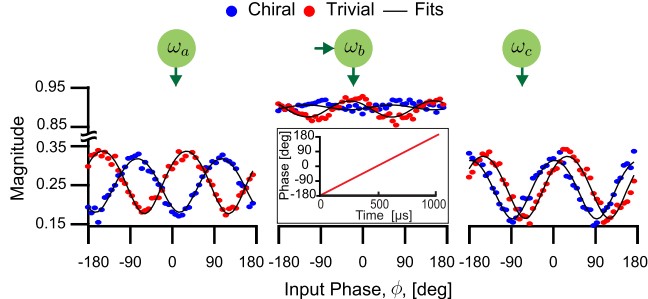

**Fig. 3 | Calibration of the gauge-invariant phase, Θ, of the 3-site chain.** The lattice cartoon depicts a signal injected in the center mode $b$ while the reflected and transported signals are measured. The input signal has a constant magnitude and a phase that sweeps from −180° to 180° during the measurement. The magnitudes of the transported signals are plotted as functions of the input phase in the trivial (red) and chiral (blue) cases, together with the theoretical fits (black). The signals are normalized such that they correspond to reflection and transport coefficients (see Methods). Note that all transported signals are strongly modulated along the input phase axis. The transported signals to modes $a$ and $c$ are in phase for the trivial chain. However, they are out of phase in the chiral case, where quadratures transported to opposite ends are orthogonal to each other.

**Fig. 2 | Twisted-tubes picture of the BKC. a** The $a$-$b$ link is represented by a pair of interleaved directional tubes, with blue (red) tubes describing transport to the left (right). For a given transport direction, the major axis of the elliptical cross section at the input (output) end determines the favored input (output) phase. At site $a$, for instance, the major axis of the red (blue) ellipse shows the favored phase transported to (from) site $b$. For the sake of visual clarity, we absorb a 180° phase change in the red tube. Curved arrows connecting the IQ planes show how a signal evolves in the IQ plane as it propagates in the corresponding direction. Importantly, the favored quadratures in the transport from $a$ to $b$ are generally not the same at the $a$ and $b$ ends; this is represented by the twisting of the tubes. The $a$ ($b$) end of the $a$-$b$ tube can be twisted by varying the difference phase, $\varphi_{ab}^-$, and sum phase, $\varphi_{ab}^+$. **b** We now consider transport in an open chain with two links $a$-$b$ and $b$-$c$ assuming $t_{ab} = t_{bc}$ and $\Delta_{ab} = \Delta_{bc}$. The interface between the two tubes at the common site $b$ determines the transport across the 3-site chain. The IQ plane of site $b$ shows misaligned tubes, where the alignment is quantified by the gauge-invariant phase $\Theta = \varphi_{ab}^+ + \varphi_{bc}^-$. **c** Maximally misaligned tubes $\Theta = 180°$ when the red tube of one link meets the blue tube of the next link. As a result, the transport along the chain is phase insensitive: a favored signal from $a$ arrives at $b$ orthogonal to the favored quadrature propagating to $c$ and is subsequently suppressed. In (**d**), by twisting the $b$ end of the $b$-$c$ tube by changing $\varphi_{bc}^-$ and aligning the tubes at $\Theta = 90°$, we create two continuous paths along the chain and realize chiral transport. We see this maximum input phase dependence (see Methods) regardless of the absolute orientation of the tubes at $b$, which is a gauge degree of freedom. In the transport from $c$ ($a$) to $a$ ($c$), the favored quadrature has the input phase $\phi = -\varphi_{ab}^+$ ($\phi = \varphi_{bc}^-$).

In a 3-site chain with two connected links, the relative orientation of the two tubes at site $b$ is quantified by the phase $\Theta = \varphi_{ab}^+ + \varphi_{bc}^-$ (Fig. 2b), which is invariant under local gauge transformations (see Methods). When the two tubes are maximally misaligned ($\Theta = 0°$ mod 180°, Fig. 2c), transport along the chain from $a$ to $c$ or from $c$ to $a$ is independent of the input phase. On the other hand, when the two tubes are exactly aligned ($\Theta = 90°$ mod 180°, Fig. 2d), we achieve the maximum transport magnitude for the favored input phase and maximum suppression of the orthogonal phase. The values $\Theta = 90°$ mod 180° and $\Theta = 0°$ mod 180° thus represent two limiting cases of the 3-site chain, which we refer to as the "chiral" chain and the "trivial" chain, respectively. In the translationally invariant BKC in Eq. (1), $\Theta$ becomes exactly $\varphi_t$ and the chiral chain is topologically nontrivial.

We calibrate the 3-site chain by injecting a signal into the central site ($b$) and observing the transport to the ends while twisting the $b$ end of the $a$–$b$ tube by varying $\varphi_{ab}^+$, effectively changing $\Theta$ (see Fig. 3). The phase is calibrated by sweeping it and comparing the complete response to theory, similar to Fig. 4. The calibrated phases are then used for subsequent measurements. Furthermore, we calibrate the magnitudes to be reflection and transport coefficients by normalizing them to an input of unit magnitude during the fitting process (see Methods). Figure 3 shows the measured phase-dependent transport for two extremal cases $\varphi_{ab}^+ = 0°$ and 90°. The approximately sinusoidal shape of the magnitudes of the transport coefficients is a manifestation of the transport sensitivity to the input phase, with the maxima (minima) corresponding to the favored (suppressed) phases. At $\varphi_{ab}^+ = 90°$ (red), the transport magnitudes to $a$ and $c$ are in phase, indicating that the same input phases are favored or suppressed. This corresponds to a trivial chain as the ends of the tubes are completely misaligned at $b$.

More interestingly, at $\varphi_{ab}^+ = 0°$ (blue), the twisted tubes are aligned and we realize a chiral chain. The experimental signature of this is that the transport magnitudes to the ends become completely out of phase. That is, when a certain input phase is enhanced in transport to $a$, its transport to $c$ is highly suppressed, and vice versa. This corresponds to the chiral chain with the ends of the two tubes aligned at $b$.

We investigate the chain chirality by performing phase-sensitive transport measurements along the length of the chain while varying $\varphi_{ab}^+$. The results, in Fig. 4, reveal the predicted chiral transport properties. The chiral regime emerges gradually as we slowly vary $\varphi_{ab}^+$ away from the trivial case $\varphi_{ab}^+ = 90°$, reaching maximum chirality at $\varphi_{ab}^+ = 0°$.

Figure 5 illustrates the line cuts corresponding to the trivial and chiral chains. In the trivial chain (red), the transported signals are nearly reciprocal and show only minimal changes in response to the varying input phase.

The constant transport magnitude can serve as a baseline for the chiral transport. Conversely, in the chiral chain (blue curves), the transport magnitudes between the ends exhibit strong sensitivity to the input phase. In Fig. 5a, for instance, the transport from the site $a$ to site $c$ is enhanced for $\phi = 0°$, compared to the baseline, while the transport of the orthogonal input phases at $\phi = -90°$ and 90° is highly suppressed. The ratios of enhanced and suppressed magnitudes, compared to the baseline, are expected to be $\frac{t+\Delta}{t-\Delta}$ and $\frac{t-\Delta}{t+\Delta}$, respectively, which give us a rough estimate of $\Delta \approx 0.4t$.

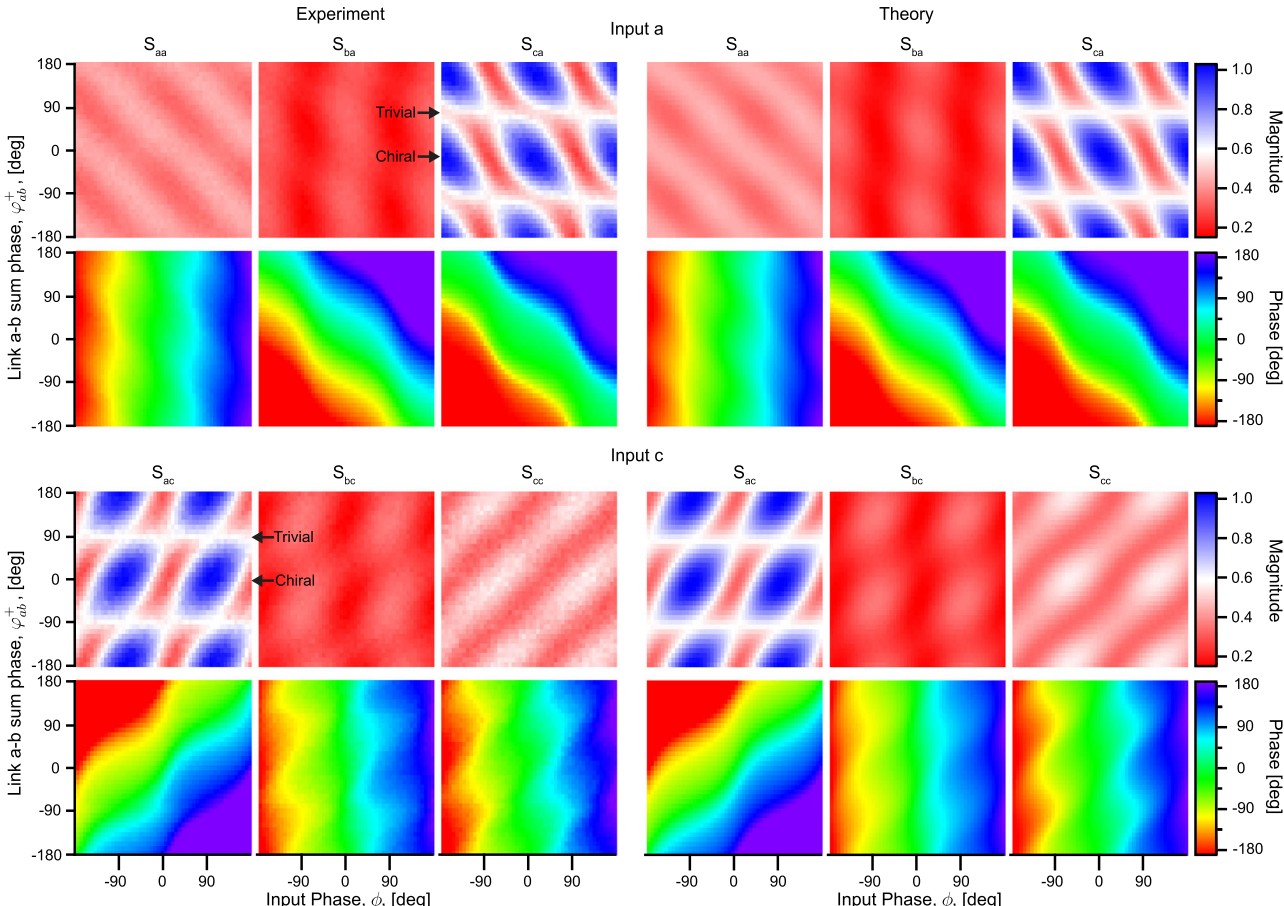

**Fig. 4 | Transport along the 3-site open chain.** The magnitude and phase of the experimental (left) and theoretical (right) normalized output signals are plotted as functions of the $a$-$b$ link sum phase, $\varphi_{ab}^+$, and input phase, $\phi$. The top panels show transport from left (site $a$) to right (site $c$), and the bottom panels show transport from right to left. The labels $\{S_{mn}\}$ indicate the output signal at site $m$ when the input signal is injected at site $n$. We clearly see that the transport between the chain ends, $S_{ac}$ and $S_{ca}$, exhibits distinct features between the trivial cases at $\varphi_{ab}^+ = \pm 90°$ and the chiral case $\varphi_{ab}^+ = 0°, \pm180°$. While the trivial transport shows little to no dependence on the input phase, the alternating blue and red regions highlight the chiral features. Figure 5 shows line cuts of the $\{S_{mn}\}$ at select link phases.

An additional signature of chirality, illustrated in Fig. 5a, is the flattening of the transport phase. Specifically, referring to the right panel of (a) and the left panel of (b), we see that in the trivial case (red), the output phase is approximately linear, the same as the input phase. Conversely in the chiral case (blue), the output phase has a stairstep shape, approximately locking to the phase of the preferred quadrature before jumping by ±180° (which is the same quadrature with opposite amplitude). We see that the observed behavior agrees well with the theoretical predictions.

We further examine the non-Hermitian topology by extracting the $x$ and $p$ quadrature wavefunctions (see Methods). The trivial and chiral chains exhibit a striking difference in the spatial support of the $x$ and $p$ wavefunctions (Fig. 5c, d). In the trivial case, the quadratures are delocalized with nearly equal weights on both ends. In stark contrast, we observe the characteristics of the NHSE in the chiral case: the $x$ and $p$ wavefunctions are strongly localized at the right and left ends, respectively. This demonstrates how we can control non-Hermitian topological effects by changing the gauge-invariant phase of our chain.

We examine the sensitivity of the chiral chain to boundary conditions by connecting the chain ends (Fig. 1d). These measurements are done on a separate device from the above measurements (see Methods). We measure the reflection coefficient around the frequency of site $b$ while varying both hopping and pairing phases of the $a$-$c$ link (Fig. 6a–c). The dominant pattern is the spectra braiding as a function of $\varphi_{ac}^t$, which is determined by the loop phase[33]. However, for certain phase conditions, we observe discontinuities in the central branch of the spectrum (Fig. 6b–d), indicating that the chain is approaching dynamical instability (see Methods).

It is remarkable that the instability is determined solely by the link phases (the pump magnitudes are constant), revealing the transition to the chiral regime of the closed chain. In the twisted-tubes picture, we start with misaligned tubes (Fig. 6e), then rotate both $a$ and $c$ of the $a$-$c$ tube until it aligns with both the $a$-$b$ and $b$-$c$ tubes, forming two directional loops (Fig. 6f). Sufficient alignment of the link phases is necessary to realize nontrivial winding, and consequently, instabilities under periodic boundary conditions (Fig. 1d).

We can give an intuitive picture of the instability in terms of circulating gain. In the absence of onsite loss, when the tubes are aligned, an initial excitation traverses the loop, being amplified indefinitely, resulting in dynamical instability. If the tubes are misaligned, the circulating signal will be amplified through one link but then deamplified through the next, allowing for a stable steady state. Local loss, as in our chain, simply shifts the eigenvalues down in the complex plane (see Fig. 1), increasing the instability threshold for $\Delta$. (The threshold is $\Delta = \frac{\kappa}{2}$). We have confirmed that for higher values of $\Delta$, the system becomes unstable, leading to coupled parametric oscillations of the modes.

## Discussion

We have demonstrated on our AQS platform that nontrivial non-Hermitian topological systems can be realized using parametric down-

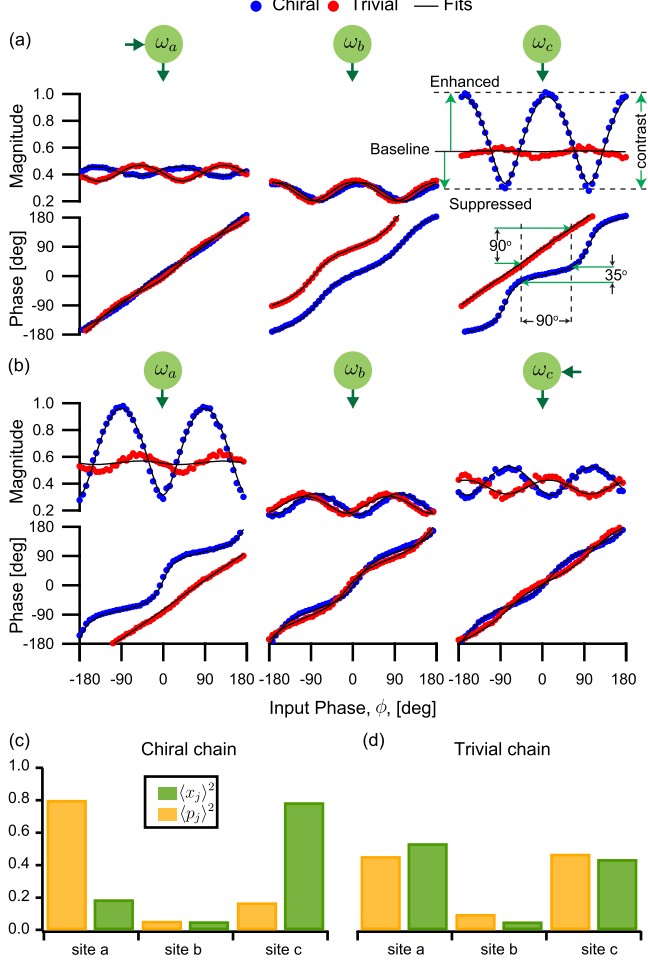

**Fig. 5 | Transport of the 3-site open chain at selected pump phases.** Line cuts of Fig. 4 are shown for the chiral chain at $\varphi_{ab}^+ = 0°$ (blue curves) and the trivial chain at $\varphi_{ab}^+ = 90°$ (red curves) as well as the fit to theory (black). **a** Signal injected at the left end. In the trivial chain, we measure largely phase-insensitive transport for all input phases. However, in the chiral chain, the magnitudes of transported signals change significantly with the input phase. We clearly note the enhancement and suppression of the transport compared to the baseline defined by the trivial chain transport. For instance, the transport is enhanced at the input phase $\phi = 0°$ and suppressed at the orthogonal phase $\phi = \pm 90°$. Moreover, the transported phase is flattened despite the fact that $\phi$ is swept in a continuous linear fashion, a strong indication of how the signal propagates through a single quadrature. **b** When a signal is injected at the right end of the chiral chain, the enhanced input phase is $\phi = \pm 90°$, which is orthogonal to the favored phase in the transport of the opposite direction. **c, d** The weights of $x$ and $p$ quadrature wavefunctions. In (**c**), the $x$ and $p$ quadrature wavefunctions in the chiral chain are localized on the opposite chain ends. In the trivial chain, however, we see in (**d**) that both wavefunctions are delocalized. In both cases, the wavefunctions have minimal support in the center site as the zero eigenmode of an odd chain is not supported on even sites.

conversion. Unlike dissipation-induced non-Hermiticity, this approach allows us to preserve some of the Hamiltonian symmetries, such as time-reversal symmetry. Therefore, this platform can be used to explore the rich topological phases and symmetries of non-Hermitian systems. Furthermore, since the dynamics are the results of a coherent process, as opposed to dissipation, in a Hermitian Hamiltonian[31], our AQS platform can implement genuine quantum dynamics with effective non-Hermiticity. An interesting future direction for our platform is to explore the interplay between interactions, topology and non-Hermiticity[49–53]. In fact, the BKC exhibits nonlinear dynamics in the above-threshold regime, where we have observed coupled parametric

oscillations in the system. This would serve as a first example of non-Hermitian non-linear quantum dynamics.

Furthermore, the 3-site chain can also be used in interesting applications. We can utilize the chiral features as a phase-dependent quantum amplifier[38,54,55]. Alternatively, vacuum squeezing can be used to realize entangled multimode states, which is a complex resource that can play a central role in continuous-variable quantum information processing, e.g., Gaussian boson sampling[56]. In addition, non-Hermitian systems have a wide range of applications from quantum sensing[57] to entanglement creation and control[58].

We discussed future directions to improve our AQS platform in earlier work[33]. Briefly, we can improve the hardware efficiency by increasing the length of the parametric cavity. This would increase the number of modes that can be utilized as lattice sites in AQS. In addition, a single parametric cavity can be used as a sublattice in a network of coupled cavities. The flexibility of controlling local coupling phases allows us to simulate topological models requiring nontrivial phase conditions, which gives our platform advantages over competing platforms.

## Methods

### Input-output theory

Here we describe in detail the input-output theory used to quantitatively study the transport in our system. Additional information on the fitting procedure is provided in Supplementary Note 4.

We denote the cavity mode $j$ as $\hat{a}_j$ and its bare frequency as $\omega_j^{(0)}$. To couple modes $j$ and $j'$, we apply a beam-splitter pump with frequency $\omega_{jj'}^t \approx |\omega_j^{(0)} - \omega_{j'}^{(0)}|$ and a down-conversion pump with frequency $\omega_{jj'}^\Delta \approx \omega_j^{(0)} + \omega_{j'}^{(0)}$; in the rotating-wave approximation, they generate the hopping and pairing terms in the Hamiltonian, respectively. More concretely, we choose a set of frequencies $\omega_j$, such that $\omega_j \approx \omega_j^{(0)}$ for all modes, and $\omega_{jj'}^t = |\omega_j - \omega_{j'}|$, $\omega_{jj'}^\Delta = \omega_j + \omega_{j'}$ for all pumps; since there are only as many free variables $\omega_j$ as the number of modes, generally the pump frequencies are not independent of each other.

We should distinguish between signal and idler frequencies in the presence of down-conversion pumps. Given the probe detuning $\Omega$, an input signal at the frequency $\omega_j + \Omega$ is coupled to other signal frequencies $\omega_{j'} + \Omega$ by beam-splitter pumps, and to other idler frequencies $\omega_{j'} - \Omega$ by down-conversion pumps. In the frequency-domain spectrum measurements, we send in a coherent state as the input signal at $\omega_j + \Omega$ and detect the reflected signal at the same frequency; the spectrum is mapped out as a function of the probe detuning $\Omega$. On the other hand, in the phase-dependent transport measurements, we set $\Omega = 0$ such that the signal and idler frequencies coincide for every mode, which generates interference between signal and idler frequencies and results in phase-dependent transport. We send in a coherent tone as the input signal at the frequency $\omega_j$ while scanning the input phase, and detect the reflected signal at $\omega_j$ and transported signal at $\omega_{j'}$.

After the rotating wave approximation, the general quadratic Hamiltonian that can be programmed in our AQS takes the following form:

$$\hat{\mathcal{H}}_S = \sum_j \hbar \delta\omega_j \hat{a}_j^\dagger \hat{a}_j + \frac{1}{2}\sum_{\langle jj'\rangle}\left(t_{jj'}e^{i\varphi_{jj'}^t}\hat{a}_j^\dagger\hat{a}_{j'} + \Delta_{jj'}e^{i\varphi_{jj'}^\Delta}\hat{a}_j\hat{a}_{j'} + \text{h.c.}\right). \quad (8)$$

Here we work in the reference frame rotating at the frequency $\omega_j$ at mode $j$, and the pump detunings are defined as $\delta\omega_j = \omega_j - \omega_j^{(0)}$. The second sum runs over all pairs of connected sites $j$ and $j'$, and the hopping and pairing strengths $t_{jj'}$ and $\Delta_{jj'}$ have tunable phases $\varphi_{jj'}^t = -\varphi_{j'j}^t$ and $\varphi_{jj'}^\Delta = \varphi_{j'j}^\Delta$, respectively.

Under the local gauge transformation $\hat{a}_j \rightarrow U\hat{a}_j U^\dagger = \hat{a}_j e^{i\theta_j}$, the link phases in the transformed Hamiltonian $\hat{\mathcal{H}}_S' = U\hat{\mathcal{H}}_S U^\dagger$ become

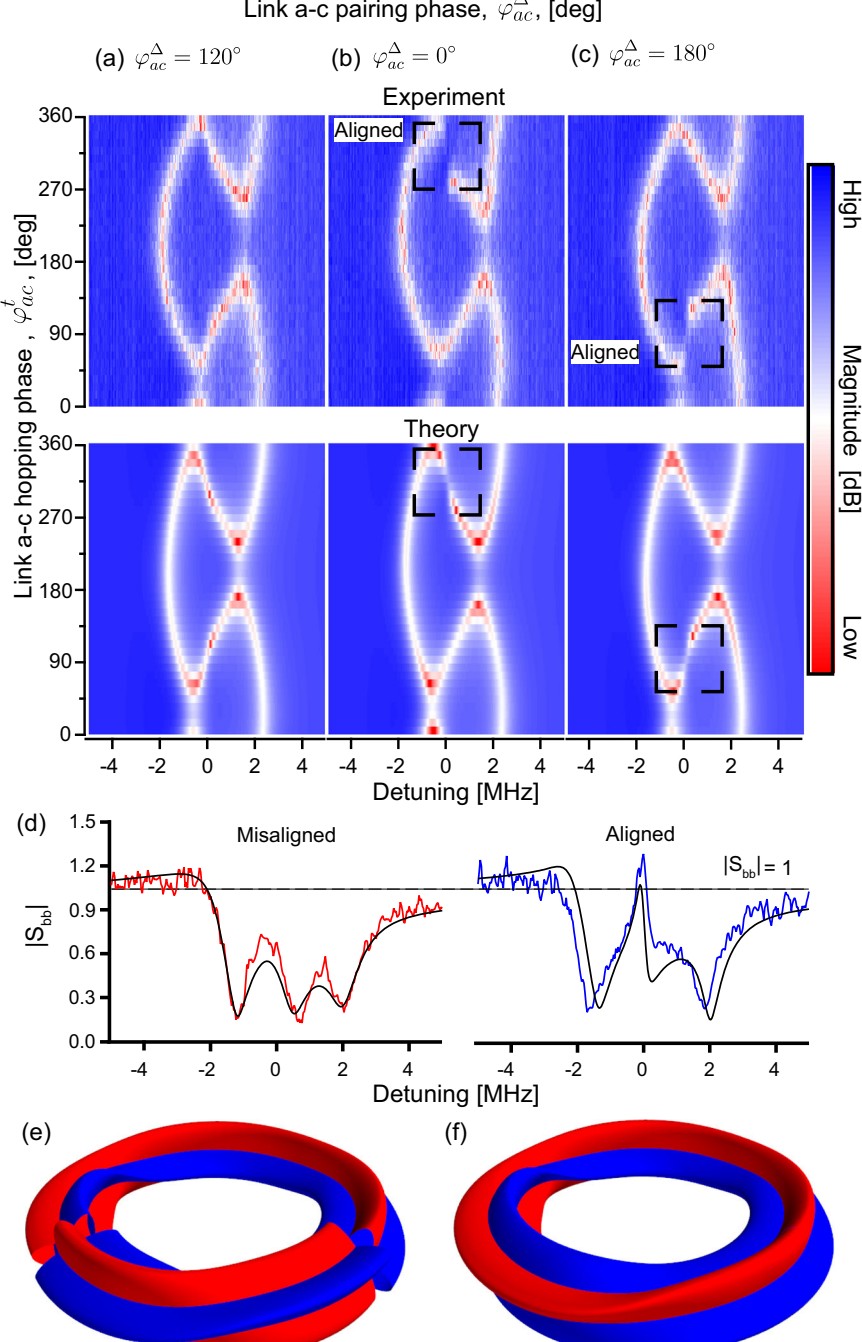

**Fig. 6 | Spectrum of the 3-site closed BKC. a–c** The measured (top) and fit (bottom) reflection magnitudes at site $b$ as functions of $\varphi_{ac}^t$ and probe detuning. From the fit, we estimate $\Delta \approx 0.41\kappa$. **a** Spectrum at $\varphi_{ac}^\Delta = 120°$. The tubes are misaligned, reducing the total gain around the loop. In this case, sweeping $\varphi_{ac}^t$ produces a braided spectrum typical of loops with only hopping. **b** At $\varphi_{ac}^\Delta = 0°$, the braided spectrum is overall similar, but we observe a discontinuity (black square) when the tubes are exactly aligned at $\varphi_{ac}^\Delta = 310°$. **c** Offsetting $\varphi_{ac}^\Delta$ by 180° forces the discontinuity to jump from the top to the bottom zero eigenmode. **d** To examine the nature of the discontinuity, we plot line cuts for the aligned (blue) and misaligned

(red) cases at a higher pairing strength ($\Delta \approx 0.68\kappa$). We see that the dip of the near-zero eigenmode turns into an amplification peak, with the reflection magnitude growing beyond 1. This indicates that the system is approaching a dynamical instability, which is the cause of the discontinuities in (**b**, **c**). We note that the magnitudes of all pumps are constant in (**a–c**), so the onset of instability is caused only by satisfying the chirality condition. (The system nonlinearity is more pronounced at this pairing strength, causing the deviation in the fit from the linear theory). **e, f** Twisted-tubes depiction of the closed chain before and after alignment.

$\varphi_{jj'}^t \to \varphi_{jj'}^t - \theta_j + \theta_{j'}$ and $\varphi_{jj'}^\Delta \to \varphi_{jj'}^\Delta + \theta_j + \theta_{j'}$. Therefore, in agreement with our twisted-tubes picture, the phases $\varphi_{jj'}^\pm = (\varphi_{jj'}^t \pm \varphi_{jj'}^\Delta)/2$ transform as $\varphi_{jj'}^+ \to \varphi_{jj'}^+ + \theta_{j'}$ and $\varphi_{jj'}^- \to \varphi_{jj'}^- - \theta_j$; furthermore, for two links $j–j'$ and $j'–j''$, the phase $\varphi_{jj'}^+ + \varphi_{j'j''}^-$ is fully gauge invariant.

Input signals are sent in through a measurement line, which is coupled to the system by the Hamiltonian

$$\hat{\mathcal{H}}_P = i \sum_j \sqrt{\kappa_j^{\text{ext}}} \left( \hat{a}_{\text{in},j}^\dagger - \hat{a}_{\text{in},j} \right) \left( \hat{a}_j^\dagger + \hat{a}_j \right). \tag{9}$$

Here $\kappa_j^{\text{ext}}$ is the external coupling rate to the input mode $\hat{a}_{\text{in},j}$. Taking the onsite single-photon loss rate $\kappa_j$ into account, we can solve the Heisenberg-Langevin equations of motion for the signal/idler modes $\hat{a}_j^{\text{S/I}}$ in terms of the corresponding input modes $\hat{a}_{\text{in},j}^{\text{S/I}}$:

$$
\begin{aligned}
&i\left[\hbar(\delta\omega_j + \Omega) + i\frac{\kappa_j}{2}\right]\langle\hat{a}_j^{\text{S}}\rangle + \sqrt{\kappa_j^{\text{ext}}}\langle\hat{a}_{\text{in},j}^{\text{S}}\rangle \\
&+ \frac{i}{2}\sum_{j'}\left(t_{jj'}e^{i\varphi_{jj'}^t}\langle\hat{a}_{j'}^{\text{S}}\rangle + \Delta_{jj'}e^{-i\varphi_{jj'}^\Delta}\langle\hat{a}_{j'}^{\text{I}\dagger}\rangle\right) = 0,
\end{aligned}
\tag{10}
$$

$$
\begin{aligned}
&-i\left[\hbar(\delta\omega_j - \Omega) - i\frac{\kappa_j}{2}\right]\langle\hat{a}_j^{\text{I}\dagger}\rangle + \sqrt{\kappa_j^{\text{ext}}}\langle\hat{a}_{\text{in},j}^{\text{I}\dagger}\rangle \\
&- \frac{i}{2}\sum_{j'}\left(t_{jj'}e^{-i\varphi_{jj'}^t}\langle\hat{a}_{j'}^{\text{I}\dagger}\rangle + \Delta_{jj'}e^{i\varphi_{jj'}^\Delta}\langle\hat{a}_{j'}^{\text{S}}\rangle\right) = 0.
\end{aligned}
\tag{11}
$$

Finally, employing the input-output relation

$$
\hat{a}_{\text{out},j}^\alpha = \sqrt{\kappa_j^{\text{ext}}}\hat{a}_j^\alpha - \hat{a}_{\text{in},j}^\alpha, \alpha = \text{S, I},
\tag{12}
$$

we express the output modes in terms of the input modes,

$$
\langle\hat{a}_{\text{out},j}^{\text{S}}\rangle = \sum_{j'}\left(S_{jj'}^{\text{SS}}\langle\hat{a}_{\text{in},j'}^{\text{S}}\rangle + S_{jj'}^{\text{SI}}\langle\hat{a}_{\text{in},j'}^{\text{I}\dagger}\rangle\right).
\tag{13}
$$

Transport properties are now given by the scattering matrix $S_{jj'}^{\alpha\beta}$. In the frequency-domain spectrum measurements, each idler frequency generally only receives a vacuum input, which is negligible compared to the coherent state input at the signal frequencies; thus the measured reflection amplitude at mode $j$ is simply $S_{jj}^{\text{SS}}$. On the other hand, in the phase-dependent transport measurements at zero probe detuning $\Omega = 0$, the signal and idler frequencies always have identical inputs. The transport from $j'$ to $j$ is therefore characterized by the normalized phase-dependent transport coefficient with unit input,

$$
S_{jj'}(\phi) = S_{jj'}^{\text{SS}}e^{i\phi} + S_{jj'}^{\text{SI}}e^{-i\phi},
\tag{14}
$$

where $\phi$ is the input phase at mode $j'$.

We extract the $x$ and $p$ quadrature wavefunctions in chiral and trivial chains in Fig. 5c, d as follows. For an input signal with phase $\phi = 0°$ at site $b$, the $x$ wavefunction at the site $j = a, b, c$ is related to the $x$ quadrature of the output mode via the input-output relation $\langle\hat{x}_j\rangle = (\langle\hat{x}_{\text{out},j}\rangle + \langle\hat{x}_{\text{in},j}\rangle)/\sqrt{\kappa_j^{\text{ext}}}$ (cf. Eq. (12)), where $\langle\hat{x}_{\text{in},j}\rangle$ is nonzero for $j = b$. To compare the different cases, we normalize the weights of the wavefunction such that $\sum_j\langle\hat{x}_j\rangle^2 = 1$. We repeat the same process to extract the $p$ wavefunctions.

## 2-mode and 3-mode open chains

In this section we consider the chiral transport in 2-mode and 3-mode BKCs with open boundary conditions in the framework of the input-output theory. For a single link $a$–$b$ with hopping $t_{ab}e^{i\varphi_{ab}^t}$ and pairing $\Delta_{ab}e^{i\varphi_{ab}^\Delta}$, solving Eqs. (10) and (11), we obtain the transport coefficient Eq. (14) from $a$ to $b$ as

$$
\begin{aligned}
&S_{ba}(\phi) \\
&= \frac{2\sqrt{\kappa_a^{\text{ext}}\kappa_b^{\text{ext}}}}{D_2}ie^{-i\varphi_{ab}^+}[t_{ab}e^{i(\phi-\varphi_{ab}^-)} + \Delta_{ab}e^{-i(\phi-\varphi_{ab}^-)}],
\end{aligned}
\tag{15}
$$

and the transport coefficient from $b$ to $a$ as

$$
\begin{aligned}
&S_{ab}(\phi) \\
&= \frac{2\sqrt{\kappa_a^{\text{ext}}\kappa_b^{\text{ext}}}}{D_2}ie^{i\varphi_{ab}^+}[t_{ab}e^{i(\phi+\varphi_{ab}^+)} + \Delta_{ab}e^{-i(\phi+\varphi_{ab}^+)}].
\end{aligned}
\tag{16}
$$

Here, the common denominator is

$$
D_2 = t_{ab}^2 - \Delta_{ab}^2 + \kappa_a\kappa_b,
\tag{17}
$$

and we define the sum and difference phases as before, $\varphi_{ab}^\pm = (\varphi_{ab}^t \pm \varphi_{ab}^\Delta)/2$. As $\Delta_{ab} \to \sqrt{t_{ab}^2 + \kappa_a\kappa_b}$, Eq. (17) indicates the transport coefficients increase rapidly and the system approaches instability, which is consistent with the upper bound of $\Delta$ in Eq. (7) for $N = 2$.

Equations (15) and (16) clearly show phase-dependent chirality, as explained in Fig. 2. The input phase $\phi = \varphi_{ab}^-$ maximizes the magnitude of Eq. (15), and corresponds to the output phase $90° - \varphi_{ab}^+$. These constitute the favored quadratures in the transport from $a$ to $b$ (major axes of the red ellipses). By contrast, the orthogonal quadratures with the input phase $\phi = 90° + \varphi_{ab}^-$ and the output phase $180° - \varphi_{ab}^+$ are suppressed (minor axes of the red ellipses). The favored quadratures from $a$ to $b$ may be different at both sites, e.g., appearing as $x$ at site $a$ but as $p$ at site $b$, $x_a \to p_b$; it is $\varphi_{ab}^-$ ($\varphi_{ab}^+$) that determines the favored quadrature at site $a$ ($b$). On the other hand, the quadratures suppressed in the transport from $a$ to $b$ are favored in the transport from $b$ to $a$ (major axes of the blue ellipses), in our example $p_a \leftarrow x_b$. By programming the coupling phases on a single link, we can tune continuously between different transport scenarios: for instance, starting from favored quadratures $x_a \to p_b$ and $p_a \leftarrow x_b$, we can vary $\varphi_{ab}^+$ to arrive at $x_a \to x_b$ and $p_a \leftarrow p_b$, or vary $\varphi_{ab}^-$ to arrive at $p_a \to p_b$ and $x_a \leftarrow x_b$.

In an open chain with two links $a$–$b$ and $b$–$c$, the transport coefficient from $b$ to $a$ is still given by Eq. (16), except that the real prefactor is replaced by $2\sqrt{\kappa_a^{\text{ext}}\kappa_b^{\text{ext}}}\kappa_c/D_3^{\text{o}}$, where

$$
D_3^{\text{o}} = (t_{ab}^2 - \Delta_{ab}^2)\kappa_c + (t_{bc}^2 - \Delta_{bc}^2)\kappa_a + \kappa_a\kappa_b\kappa_c.
\tag{18}
$$

Similarly, making the substitutions $b \to c$, $a \to b$ and replacing the real prefactor with $2\sqrt{\kappa_b^{\text{ext}}\kappa_c^{\text{ext}}}\kappa_a/D_3^{\text{o}}$ in Eq. (15), we obtain the transport coefficient from $b$ to $c$. As discussed in the main text, the favored quadrature from $b$ to $a$ is $\phi = -\varphi_{ab}^+$ while that from $b$ to $c$ is $\phi = \varphi_{bc}^-$, and the difference between the two is given by the gauge-invariant phase $\Theta = \varphi_{ab}^+ + \varphi_{bc}^-$. We also mention that Eq. (18) is consistent with Eq. (7) for $N = 3$.

The phase dependence of the transport coefficient from $a$ to $c$ is slightly more complicated:

$$
\begin{aligned}
&S_{ca}(\phi) \\
&= \frac{2\sqrt{\kappa_a^{\text{ext}}\kappa_c^{\text{ext}}}}{D_3^{\text{o}}}e^{-i\varphi_{bc}^+}\big[(-t_{ab}t_{bc}e^{-i\Theta} + \Delta_{ab}\Delta_{bc}e^{i\Theta})e^{i(\phi-\varphi_{ab}^-)} \\
&+ (-\Delta_{ab}t_{bc}e^{-i\Theta} + t_{ab}\Delta_{bc}e^{i\Theta})e^{-i(\phi-\varphi_{ab}^-)}\big].
\end{aligned}
\tag{19}
$$

While $\varphi_{ab}^-$ ($\varphi_{bc}^+$) determines the favored quadrature at site $a$ ($c$), the other two link phases enter Eq. (19) only through the gauge-invariant linear combination $\Theta$. When $\Theta = \pm 90°$ ($\Theta = 0°$ mod $180°$), a constructive (destructive) interference ensues, and we find a strong (weak) dependence on the input phase $\phi$. All the above points are consistent with the twisted tubes picture.

## 3-mode closed chain

In this section we study how a closed 3-mode chain can approach dynamical instability when we tune its link phases. Here we choose to examine the determinant of the coefficient matrix of the 6 coupled Heisenberg-Langevin equations, Eqs. (10) and (11), which come from the input-output theory; solving an eigenvalue problem will lead to the same quantity. We focus on vanishing pump detuning $\Omega = 0$, where discontinuities are seen to arise in the spectrum in Fig. 6. For a closed

chain of three links, $a$–$b$, $b$–$c$ and $c$–$a$, the determinant is found as

$$D_3^c = \left\{ 4 t_{ab} t_{ca} \Delta_{ab} \Delta_{ca} \left[ (t_{bc}^2 + \Delta_{bc}^2) \cos 2\Theta_a \right.\right.$$
$$\left. - t_{bc}^2 \cos 2(\Theta_b + \Theta_c) - \Delta_{bc}^2 \cos 2(\Theta_b - \Theta_c) \right]$$
$$+ 2 t_{bc}^2 \Delta_{ca}^2 \Delta_{ab}^2 \cos 2(\Theta_b + \Theta_c - \Theta_a) + \text{perm.} \Big\}$$
$$+ 2 t_{bc}^2 t_{ca}^2 t_{ab}^2 \cos 2(\Theta_a + \Theta_b + \Theta_c) + C_3^c, \tag{20}$$

where we have defined the gauge-invariant combinations $\Theta_a = \varphi_{ca}^+ + \varphi_{ab}^-$, $\Theta_b = \varphi_{ab}^+ + \varphi_{bc}^-$ and $\Theta_c = \varphi_{bc}^+ + \varphi_{ca}^-$, and "perm." represents the permutation-symmetric contributions $(abc) \to (bca), (cab)$. The last term $C_3^c$ is independent of phases,

$$C_3^c = K_3 + 2 t_{bc}^2 t_{ca}^2 t_{ab}^2 - 4 \Delta_{ab}^2 \Delta_{bc}^2 \Delta_{ca}^2$$
$$+ (2 t_{bc}^2 \Delta_{ab}^2 \Delta_{ca}^2 - 4 \Delta_{bc}^2 t_{ab}^2 t_{ca}^2 + \text{perm.}),$$
$$K_3 = \left[ \kappa_a \kappa_b \kappa_c + \kappa_a (t_{bc}^2 - \Delta_{bc}^2) + \kappa_b (t_{ca}^2 - \Delta_{ca}^2) \right.$$
$$\left. + \kappa_c (t_{ab}^2 - \Delta_{ab}^2) \right]^2. \tag{21}$$

Note that the link phases appear exclusively through the three gauge-invariant combinations. In the open chain limit of $t_{bc} = \Delta_{bc} = 0$, we can explicitly verify that all phase dependence drops out. Also, if all pairing strengths vanish, only the loop phase $\Theta_a + \Theta_b + \Theta_c$ remains.

When all pairing terms are turned off, simple algebra shows that $D_3^c$ is positive definite as a function of pump phases. Once the pairing strength exceeds a threshold value, $D_3^c$ can become zero for some link phases and Eqs. (10) and (11) become singular: this is the point where the closed chain turns unstable and the linear theory fails. This motivates us to find the minimum of $D_3^c$ with respect to all link phases.

Equation (20) allows an analytical minimization with respect to $\Theta_a$ by expanding out the cosine and sine terms,

$$D_3^c \geq 8 t_{ab} t_{bc} \Delta_{ab} \Delta_{bc} (t_{ca}^2 + \Delta_{ca}^2) \cos 2\Theta_b$$
$$+ 8 t_{ca} t_{bc} \Delta_{ca} \Delta_{bc} (t_{ab}^2 + \Delta_{ab}^2) \cos 2\Theta_c$$
$$- 8 t_{ab} t_{ca} \Delta_{ab} \Delta_{ca} \left[ t_{bc}^2 \cos 2(\Theta_b + \Theta_c) \right.$$
$$\left. + \Delta_{bc}^2 \cos 2(\Theta_b - \Theta_c) \right] + \text{const.} \tag{22}$$

Since cosines range between −1 and 1, it is clear that the minimum with respect to $\Theta_b$ and $\Theta_c$ is found at $\cos 2\Theta_b = \cos 2\Theta_c = -1$. Consistent with the permutation symmetry, at this point we also have $\cos 2\Theta_a = -1$. This is reminiscent of the chiral transport regime in the open 3-mode system, achieved at $\Theta = \pm 90°$. The minimum of $D_3^c$ with respect to link phases reads

$$D_{3,\text{min}}^c = K_3 - 4 \left( \Delta_{bc} \Delta_{ca} \Delta_{ab} + \Delta_{bc} t_{ab} t_{ca} + \Delta_{bc} t_{ca} \Delta_{ab} \right.$$
$$\left. + \Delta_{bc} \Delta_{ca} t_{ab} \right)^2. \tag{23}$$

In particular, when all hopping/pairing terms have the same strength $t_{bc} = t_{ca} = t_{ab} = t$ and $\Delta_{bc} = \Delta_{ca} = \Delta_{ab} = \Delta$ and all loss rates are equal $\kappa_a = \kappa_b = \kappa_c = \kappa$,

$$D_{3,\text{min}}^c = (\kappa^2 - 4\Delta^2)[3 t^2 + (\kappa + \Delta)^2][3 t^2 + (\kappa - \Delta)^2]. \tag{24}$$

Therefore, if the pairing strength satisfies $\Delta > \kappa/2$, the closed chain can become unstable for some choices of link phases. This is consistent with the stability/instability condition Eq. (7).

Following the procedure in the main text, we align the $a$–$b$ and $b$–$c$ tubes at $b$ by fixing $\cos 2\Theta_b = -1$. Consequently

$$D_3^c = -2 \left[ t_{ca} (t_{ab} t_{bc} + \Delta_{ab} \Delta_{bc}) \cos(\varphi_{ac}^t - \varphi_{ab}^- - \varphi_{bc}^+) \right.$$
$$\left. - \Delta_{ca} (t_{ab} \Delta_{bc} + \Delta_{ab} t_{bc}) \cos(\varphi_{ac}^\Delta + \varphi_{ab}^- - \varphi_{bc}^+) \right]^2$$
$$+ K_3 + 4(t_{bc}^2 - \Delta_{bc}^2)(t_{ca}^2 - \Delta_{ca}^2)(t_{ab}^2 - \Delta_{ab}^2), \tag{25}$$

where we have used $\Theta_c + \Theta_a = -\varphi_{ac}^t + \varphi_{ab}^- + \varphi_{bc}^+$ and $\Theta_c - \Theta_a = -\varphi_{ac}^\Delta - \varphi_{ab}^- + \varphi_{bc}^+$. Equation (25) indicates that provided $\cos 2\Theta_b = -1$, $D_3^c$ has two nonequivalent minima in the $\varphi_{ac}^t - \varphi_{ac}^\Delta$ plane, found at $\cos(\varphi_{ac}^t - \varphi_{ab}^- - \varphi_{bc}^+) = -\cos(\varphi_{ac}^\Delta + \varphi_{ab}^- - \varphi_{bc}^+) = \pm 1$. This agrees with Fig. 6b, c, which shows both $\varphi_{ac}^t$ and $\varphi_{ac}^\Delta$ change by 180° between the two discontinuities in the spectrum.

## Calibration and characterization

An arbitrary chain is calibrated by activating each link separately while turning off the remaining links. First, upon activating the hopping term of the link $j$–$j'$, the single mode resonance splits into two resonances whose frequency difference gives twice the coupling strength, $2 t_{jj'}$. We choose $t_{jj'}$ to be in the strong-coupling regime: the resonance splitting is greater than photon decay rates, $2 t_{jj'} > \kappa_j, \kappa_{j'}$, such that the split resonances are resolved. Furthermore, we set $t_{jj'}$ to be roughly equal along the chain.

Then, we activate each link's pairing term. In this case, there is no simple spectral feature that quantifies the pairing strength, $\Delta_{jj'}$. We roughly calibrate $\Delta_{jj'}$ using transport measurements in the following way. With both hopping and pairing terms applied, we sweep the input phase of the signal described above at site $j$ and measure the contrast of the transport at site $j'$ as a function of input phase. Here the contrast, defined as $(A_{\max} - A_{\min})/(A_{\max} + A_{\min})$ where $A_{\max}$ ($A_{\min}$) is the maximum (minimum) magnitude of the transported signal, evaluates to $\min\{t, \Delta\} / \max\{t, \Delta\}$ according to Eqs. (15) and (16). We then vary the pairing pump power, interpreting the power where the observed contrast is maximum as $\Delta_{jj'} \approx t_{jj'}$. As we further increase the pairing pump power, the system eventually becomes dynamically unstable, as discussed below Eq. (17). We choose a pairing strength that satisfies the stability/instability condition Eq. (7) for $N = 3$; this means the 3-site open chain is dynamically stable, and the spectrum of the 3-chain closed chain shows a discontinuity at some pump phases (Fig. 6).

The phases of the signals acquire arbitrary offsets from traveling through the measurement lines between the instrument and the sample. We calibrate the input and output phases of the system as follows. First, we connect the 3-site chain with hopping and pairing terms, and perform phase-sensitive transport measurements when a signal is injected in the center site $b$. We then tune the chain gauge-invariant phase $\Theta$ until we realize the chiral chain. To conveniently present the results, we finally set the right-moving quadratures as $I$ and the left-moving quadratures as $Q$ in the chiral chain.

## Data availability

The data that support the findings of this study are available from the corresponding author upon request.

## Code availability

The code used for data analysis and simulation are available from the corresponding author upon request. Because the analysis code is closely integrated with instrument-control code, which depends on specialized instrument drivers, it is not meaningful to share the code in a general repository.

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

## Acknowledgements

The authors wish to thank B. Plourde, J.J. Nelson and M. Hutchings at Syracuse University for invaluable help in junction fabrication for the device used to produce Fig. 6. C.M.W., J.H.B., Z.S., D.D., I.N., J.S.C.H., and C.W.S.C. acknowledge the Canada First Research Excellence Fund (CFREF), NSERC of Canada, the Canadian Foundation for Innovation, the Ontario Ministry of Research and Innovation, and Industry Canada for financial support. Device fabrication was done at the University of Waterloo's QNFCF facility. This infrastructure would not be possible without the significant contributions of CFREF-TQT, CFI, ISED, the OMRI, and Mike & Ophelia Lazaridis. A.A.C., A.M. acknowledge support from the Air Force Office of Scientific Research under Grant No. FA9550-19-1-0362, and the Simons Foundation through a Simons Investigator Award (Grant No. 669487, A.A.C.).

## Author contributions

J.H.B., D.D., and C.M.W. contributed to the experimental work. J.S.C.H. contributed to the experimental work while a student at the University of Waterloo, prior to joining AWS Center for Quantum Computing. I.N., D.D., and C.W.S.C. fabricated the devices. Z.S., A.M., and A.A.C contributed to the theoretical work.

## Competing interests

The authors declare no competing interests.
