## [Peer Review File · Nature Communications]

Quantum Simulation of the Bosonic Kitaev ChainREVIEWER COMMENTS

Reviewer #1 (Remarks to the Author):

The authors present an experimental realization of the bosonic Kitaev chain model using parametric cavities. The needed hopping and pairing terms are implemented with difference and sum of the mode frequencies. They demonstrate important precursors of nontrivial topology and the non-Hermitian skin effect in the bosonic Kitaev chain, including chiral transport, quadrature wavefunction localization, and sensitivity to boundary conditions. While the work is well done, sound and very interesting, and significant for the many-body topological quantum physics with artificial systems.

I have the following comments before recommending its publication.

First, the authors claimed to implement the bosonic Kitaev chain in synthetic dimensions. The bosonic modes are in different cavities with different sites, and thus “synthetic dimensions” may be misleading.

Second, I suggest to include previous work on chiral transport with parametric couplings on superconducting circuits, e.g., *Sci. Rep.* 5, 8352 (2015) and its realization in *Nat. Phys.* 13, 146 (2017), and comment on the difference with here.

Reviewer #2 (Remarks to the Author):

The authors describe a method for synthesizing a Bosonic Kitaev Chain using a coplanar waveguide resonator grounded through a SQUID. The SQUID can be parametrically modulated at both difference and sum frequencies in order to generate interactions between multiple resonant modes (“bosonic sites or nodes”) of the resonator. In the case of two and three linked bosonic sites, the authors use microwave transport measurements at the different nodes to check their predictions. Because each node is coupled to the feedline for driving and extraction of microwave signals, this open quantum system is accessible to characterization. Under the right conditions (see Eqn. (7)), even though the pump drives at the sum frequencies continuously push energy into the system, the system can reach a

steady state with energy leaking out of the connected nodes with rates κ_j . The authors describe a graphical method for visualizing the stable dynamics in the form of “twisted-tubes”. The presentation of the data and theory are very clear and seem to agree extremely well.

I find several noteworthy results. First, the presentation of the data and the visualization are very clear. The ability to generate both hopping and pairing terms simultaneously with tunable strengths is also a unique part of this platform. I also find the Methods thorough and appreciated the detail in D. and E. describing the calibrations and fitting procedures, as well as the Tables with device parameters.

I think this work will be of significance to the field of quantum simulation. It represents a rather simple system to fabricate and study, but with rich dynamics built in. It can help to motivate more work and push towards more complex systems. It adds a novel experimental existence proof of a fully tunable analog simulator to the current literature for this nascent field that is still maturing experimentally.

I feel that the work supports the conclusions and claims as evidenced by the good agreement between input-output theoretical predictions and the experiment. The methodology appears to be sound and meets the expected standards in field.

I feel this work should be published in Nat. Comm., but I have a few suggestions and recommendations that I believe would improve the paper:

1) It would be nice to see in the Methods some more experimental details about the actual devices, fabrication, and the wiring diagram, including some details about the instrumentation used to record the data. This would give any reader more information were they interested in reproducing or extending results like this.

2) Although the concluding paragraphs discussed future work, there was no mention of how difficult it would be to add more nodes to this existing system. It seems that adding even one more node would not have been that difficult, since those modes are already in the current structure. What prevented this? If the authors wanted to scale this system, how many experimentally accessible mode frequencies are reasonably accessible? Does adding many more pumps complicate the system?

3) You should add the reference: S. de Léséleuc et al., Science 10.1126/science.aav9105

(2019)

Reviewer #3 (Remarks to the Author):

Busnaina et al. report the experimental realization of the minimal instance of the bosonic Kitaev chain (BKC), a lattice model with nearest neighbor complex hopping and pairing terms. The lattice sites are represented by frequency modes of a multimode superconducting cavity with hopping and pairing interactions induced by external coherent pumps. This allows the authors to control both the magnitude and phase of each coherent coupling term individually and thereby reach a suitable parameter regime and boundary conditions where the system becomes non-reciprocal and can show effects related to non-trivial topology (but maybe not a non-trivial topological phase as stated below). In particular, by performing phase-sensitive transport experiments, the authors demonstrate chiral (non-reciprocal) transport in the system, wavefunction localization, and high sensitivity of the spectrum to boundary conditions, known as the non-Hermitian skin effect.

I think that the experimental realization of lattice models showing precursors of non-trivial non-Hermitian topology can be of great importance for the field of Quantum Simulation and Condensed Matter. There are many works in the literature discussing theoretical aspects of topology in non-Hermitian systems, but few have shown direct experimental consequences. In particular, the reported realization of the BKC is interesting, which is possible due to the high level of control and programmability demonstrated by the authors in their superconducting lattice simulator. The possibility of individually controlling the amplitude of phases of all couplings as well as the switching from open and periodic boundary conditions is remarkable, even for a minimal instance of the model with only three sites. The experimental demonstration of non-reciprocal transport, localization of the wavefunctions, and the non-Hermitian skin effect via comparing eigen-spectra for open and periodic boundaries is scientifically sound and interesting. The experimental evidence and details of these results are convincing and allow reproducibility. So, from the experimental perspective, I see this work as solid.

However, I have important doubts about the theoretical foundations of various strong

claims found in the paper regarding the “non-trivial topology of the model”, the role played by dissipation, and the “genuine quantum conditions” of the experiment. Below I explain my concerns, which I require the authors to carefully clarify so that I can assess if the results of the work are well communicated and support the claims, or if some of them need to be revised and explained more precisely. Depending on this I can give a final recommendation for publication of the paper in Nature Communications.

1) My most important concern has to do with the proper justification of the non-trivial topology presented by the BKC. The authors show that the model features non-reciprocity, localized wavefunctions on the edges, and non-Hermitian skin effect, but the phase of the system can still be topologically trivial if there is no underlying topological invariant (see for instance Guo et al., arXiv:2304.06926). If this is the case, then the properties of the phase do not rely on a truly non-local topological property of the system, and the response of the system to disorder is not exponentially suppressed or protected. To be sure that the system is in a non-trivial topological phase, topological invariants need to be calculated such as the winding number calculated in Gomez-Leon et al., Quantum 7, 1016 (2023). Here, the same BKC model is considered with $\phi = \pi/2$, but also local squeezing terms $a_j^\dagger a_j^\dagger$ are included in addition to the non-local ones $a_j^\dagger a_{j+1}^\dagger$. It is shown that local and non-local squeezing terms are required in order to have a non-zero winding number $W=1$. When local pairing terms are absent, it is also shown that there appear localized edge states, but still, the phase is topologically trivial with winding number zero. I strongly recommend the authors calculate the winding number corresponding to their model and depending on this revise their claims on the non-trivial topology of the model, which may be misleading at the moment. Another important evidence of non-trivial topology would be the robustness to disorder of the chiral transport and localization of the wavefunctions, which is also absent in the discussion so far. This can be a clear indicator of topological protection, even if the lattice has few sites.

2) Another important point I ask the authors to clarify is the claim that the system can show non-Hermitian topology “even in the absence of dissipation”. I agree that the presence of pairing terms leads to a dynamical matrix that is non-Hermitian, but as far as I see from the theoretical description in Eqs. (4), (6) and (7), if local dissipation κ vanishes, then the

resulting phase is always unstable. It is very important that the authors clarify this further since stability is an unavoidable physical requirement, and therefore would make dissipation as essential as pairing terms of the realization of the BKC model. In addition, I note that all experimental data is shown for κ on the order of the hopping t so this also does not help to make this point clearer. Finally, a small local κ will be always present in the model due to practical reasons for manipulating the system via external transmission lines (whose couplings contribute to the dissipation terms). I suggest the authors to comment on all these aspects.

3) My third concern has to do with the statements “genuinely quantum simulation” or “genuinely quantum conditions” found in several places in the text. I wonder to what extent the presented simulation of the model is quantum. As far I understand the authors send and measure coherent (Gaussian fields) which can be equally well described by a classical model of linearly coupled modes such as in Naaman & Aumentado, PRX Quantum 3, 020201 (2022). I think the key is to measure quantum fluctuations in the system, of the generated squeezed states, for instance. I think this may be possible in the discussed setup, but I do not see clearly to what extent this is reached in the present work. I, therefore, ask the authors to be clearer in this respect and to soften their claims regarding the quantum regime if this is needed.

4) Another important comment is regarding the various approaches to non-Hermitian topology highlighted in the introduction. Only two options are mentioned: either take an open quantum system and post-select the dynamics condition to the absence of jumps or consider a Hermitian system with squeezing terms as described in the present work. Here, the authors are missing a relevant approach, which is considering a general open quantum system (with not only pairing terms but also non-local pumping or dissipation) and analyzing its non-trivial topology by mapping to topological insulator theory via the singular value decomposition and the doubled Hamiltonian [Gong et al. PRX 8, 031079 (2018), Porras et al. PRL 122, 143901, (2019), Herviou, et al. Phys. Rev. A 99, 052118 (2019), Okuma et al. Phys. Rev. B 102, 014203 (2020)]. This approach includes the non-Hermiticity via pairing terms as a special case, and it is more general as it allows the classification of non-trivial topological phases according to the underlying symmetries.

5) Finally, when commenting on applications of the system to topological amplification, the list does not include relevant references in the field such as Wanjura et al., Nat. Commun. 11, 3149 (2020) and Ramos et al. PRA 103, 033513 (2021).

Reply to Referees

To the Referees,

We would like to thank the Referees for their time and careful consideration of our manuscript. The comments and criticism of the Referees, which we found to be largely positive, have helped us to refine and improve the manuscript.

In the following parts of this letter, we have made an effort to respond to all of the referees comments. The comments of the referees are reproduced in the shaded boxes. Our replies are in the regular text. Within our replies, the blue text summarizes changes and additions to our manuscript. In the updated manuscript file, major changes and additions are in red text. Finally, as part of our reply we have added a new Supplementary Information (SI) section. This new SI section includes, among other things, the additional experimental details requested by Referee #2 as well as a more detailed discussion of topological invariant of the model, as requested by Referee #3. We also include some additional numerics demonstrating the robustness of the model to disorder.

We hope that with the changes and additions the Referees find the updated manuscript ready for publication in Nature Communications.

Response to Referee #1

The authors present an experimental realization of the bosonic Kitaev chain model using parametric cavities. The needed hopping and pairing terms are implemented with difference and sum of the mode frequencies. They demonstrate important precursors of nontrivial topology and the non-Hermitian skin effect in the bosonic Kitaev chain, including chiral transport, quadrature wavefunction localization, and sensitivity to boundary conditions. While the work is well done, sound and very interesting, and significant for the many-body topological quantum physics with artificial systems.

I have the following comments before recommend its publication.

We thank the Referee for their positive evaluation of our manuscript and their helpful suggestions.

First, the authors claimed to implement the bosonic Kitaev chain in synthetic dimensions. The bosonic modes are in different cavities with different site, and thus “synthetic dimensions” may be misleading.

This is a misunderstanding. The bosonic Kitaev chain is implemented in a single cavity using multiple modes with different resonance frequencies. We believe that this use of “synthetic dimension”, in this case frequency, to denote using internal degrees of freedom of a single object is consistent with the literature.

Second, I suggest to include previous work on chiral transport with parametric couplings on superconducting circuits, e.g., Sci. Rep. 5, 8352 (2015) and its realization in Nat. Phys. 13, 146 (2017), and comment on the difference with here.

We thank the Referee for pointing to these references. They are now cited in the Introduction section of the main text. As discussed there, the main difference from our work is that our work also includes parametrically activated pairing

terms which are crucial in realizing the effective non-Hermiticity; in addition, all degrees of freedom are located in a single multimode cavity in our work, increasing the hardware efficiency.

In the revised manuscript, we have added the following sentence to the Introduction section: “We note similar work has been done in the context of classical non-Hermitian optics [1, 2] and chiral photon transport [3, 4].”

Response to Referee #2

The authors describe a method for synthesizing a Bosonic Kitaev Chain using a coplanar waveguide resonator grounded through a SQUID. The SQUID can be parametrically modulated at both difference and sum frequencies in order to generate interactions between multiple resonant modes (“bosonic sites or nodes”) of the resonator. In the case of two and three linked bosonic sites, the authors use microwave transport measurements at the different nodes to check their predictions. Because each node is coupled to the feedline for driving and extraction of microwave signals, this open quantum system is accessible to characterization. Under the right conditions (see Eqn. (7)), even though the pump drives at the sum frequencies continuously push energy into the system, the system can reach a steady state with energy leaking out of the connected nodes with rates κ_j . The authors describe a graphical method for visualizing the stable dynamics in the form of “twisted-tubes”. The presentation of the data and theory are very clear and seem to agree extremely well.

I find several noteworthy results. First, the presentation of the data and the visualization are very clear. The ability to generate both hopping and pairing terms simultaneously with tunable strengths is also a unique part of this platform. I also find the Methods thorough and appreciated the detail in D. and E. describing the calibrations and fitting procedures, as well as the Tables with device parameters.

I think this work will be of significance to the field of quantum simulation. It represents a rather simple system to fabricate and study, but with rich dynamics built in. It can help to motivate more work and push towards more complex systems. It adds a novel experimental existence proof of a fully tunable analog simulator to the current literature for this nascent field that is still maturing experimentally.

I feel that the work supports the conclusions and claims as evidenced by the good agreement between input-output theoretical predictions and the experiment. The methodology appears to be sound and meets the expected standards in field.

I feel this work should be published in Nat. Comm., but I have a few suggestions and recommendations that I believe would improve the paper:

We thank the Referee for their positive evaluation of our manuscript and their helpful suggestions and recommendations.

1) It would be nice to see in the Methods some more experimental details about the actual devices, fabrication, and the wiring diagram, including some details about the instrumentation used to record the data. This would give any reader more information were they interested in reproducing or extending results like this.

We have now included the additional experimental details requested by the Referee in the Supplementary Information.

2) Although the concluding paragraphs discussed future work, there was no mention of how difficult it would be to add more nodes to this existing system. It seems that adding even one more node would not have been that difficult, since those modes are already in the current structure. What prevented this? If the authors wanted to scale this system, how many experimentally accessible mode frequencies are reasonably accessible? Does adding many more pumps complicate the system?

In principle, it is experimentally straightforward to add more nodes to the existing system. There are 13 modes within the measurement bandwidth of 4–12 GHz, which can be readily incorporated into the chain with more flux pumps without frequency crowding. However, the presented experiment already used eleven microwave sources, which is the limit of our current instrument capacity. In more detail, a 3-mode closed chain requires six microwave sources as pumps. We also use two microwave sources in upconversion and downconversion circuits and three microwave

sources as LOs for RF digitizers used to acquire data. We note, however, that the resources required for this type of purpose-built quantum simulation circuit is still less than would be required for a gate-based simulation on a universal quantum computer.

As part of future work, we will also explore alternative ways to generate the large number of pumps required to scale the system, for instance using direct-digital synthesis of multiple tones from a single high-speed digital-to-analog converter.

We have added this discussion in the Supplementary Information.

3) You should add the reference: S. de Léséleuc et al., Science 10.1126/science.aav9105 (2019)

We thank the Referee for bringing this reference to our attention. It is now cited in the Discussion section in the main text.

In the revised manuscript, in the Discussion section, we modified the sentence:

“In addition, the model exhibits nonlinear dynamics in the above-threshold regime. For instance, we have observed coupled parametric oscillations in the system, which can be interesting to explore in future work.”

to read:

“An interesting future direction for our platform is to explore the interplay between interactions, topology and non-Hermiticity [5–8]. In fact, the BKC exhibits nonlinear dynamics in the above-threshold regime, where we have observed coupled parametric oscillations in the system.”

Response to Referee #3

Busnaina et al. report the experimental realization of the minimal instance of the bosonic Kitaev chain (BKC), a lattice model with nearest neighbor complex hopping and pairing terms. The lattice sites are represented by frequency modes of a multimode superconducting cavity with hopping and pairing interactions induced by external coherent pumps. This allows the authors to control both the magnitude and phase of each coherent coupling term individually and thereby reach a suitable parameter regime and boundary conditions where the system becomes non-reciprocal and can show effects related to non-trivial topology (but maybe not a non-trivial topological phase as stated below). In particular, by performing phase-sensitive transport experiments, the authors demonstrate chiral (non-reciprocal) transport in the system, wavefunction localization, and high sensitivity of the spectrum to boundary conditions, known as the non-Hermitian skin effect.

I think that the experimental realization of lattice models showing precursors of non-trivial non-Hermitian topology can be of great importance for the field of Quantum Simulation and Condensed Matter. There are many works in the literature discussing theoretical aspects of topology in non-Hermitian systems, but few have shown direct experimental consequences. In particular, the reported realization of the BKC is interesting, which is possible due to the high level of control and programmability demonstrated by the authors in their superconducting lattice simulator. The possibility of individually controlling the amplitude of phases of all couplings as well as the switching from open and periodic boundary conditions is remarkable, even for a minimal instance of the model with only three sites. The experimental demonstration of non-reciprocal transport, localization of the wavefunctions, and the non-Hermitian skin effect via comparing eigen-spectra for open and periodic boundaries is scientifically sound and interesting. The experimental evidence and details of these results are convincing and allow reproducibility. So, from the experimental perspective, I see this work as solid.

However, I have important doubts about the theoretical foundations of various strong claims found in the paper regarding the “non-trivial topology of the model”, the role played by dissipation, and the “genuine quantum conditions” of the experiment. Below I explain my concerns, which I require the authors to carefully clarify so that I can assess if the results of the work are well communicated and support the claims, or if some of them need to be revised and explained more precisely. Depending on this I can give a final recommendation for publication of the paper in Nature Communications.

We thank the Referee for a careful reading of our manuscript, and for their compliments regarding the experimental capabilities of our setup.

1) My most important concern has to do with the proper justification of the non-trivial topology presented by the BKC. The authors show that the model features non-reciprocity, localized wavefunctions on the edges, and non-Hermitian skin effect, but the phase of the system can still be topologically trivial if there is no underlying topological invariant (see for instance Guo et al., arXiv:2304.06926). If this is the case, then the properties of the phase do not rely on a truly non-local topological property of the system, and the response of the system to disorder is not exponentially suppressed or protected. To be sure that the system is in a non-trivial topological phase, topological invariants need to be calculated such as the winding number calculated in Gomez-Leon et al., Quantum 7, 1016 (2023). Here, the same BKC model is considered with $\phi = \pi/2$, but also local squeezing terms $a_j^\dagger a_j^\dagger$ are included in addition to the non-local ones $a_j^\dagger a_{j+1}^\dagger$. It is shown that local and non-local squeezing terms are required in order to have a non-zero winding number $W = 1$. When local pairing terms are absent, it is also shown that there appear localized edge states, but still, the phase is topologically trivial with winding number zero. I strongly recommend the authors calculate the winding number corresponding to their model and depending on this revise their claims on the non-trivial topology of the model, which may be misleading at the moment. Another important evidence of non-trivial topology would be the robustness to disorder of the chiral transport and localization of the wavefunctions, which is also absent in the discussion so far. This can be a clear indicator of topological protection, even if the lattice has few sites.

The Referee brings up a good point which is supported by both Ref. [9] and their reference Ref. [10]: the absence or presence of localized skin modes is not enough to determine whether or not a non-Hermitian system is topologically

non-trivial. To show that this is the case, one must compute relevant topological invariants or show an explicit insensitivity to disorder.

There are, however, several different notions of nontrivial topology depending on the underlying symmetry of the model. Consequently, there are several different topological invariants that one can define and compute [11]. The second reference provided by the Referee Ref. [12] presents one such invariant, in this instance a winding number. Since the k -space non-Hermitian Hamiltonian of interest in the ideal bosonic Kitaev chain (BKC) model is [13]

$$\sigma_z \mathbf{h}_B(k) = \begin{pmatrix} t \sin k & i\Delta \cos k \\ i\Delta \cos k & t \sin k \end{pmatrix} \quad (1)$$

we can immediately obtain the eigenvalues of Eq. (1) and from Eq. 33 in Ref. [12], we can then write the topological invariant defined there as

$$W_1(\omega) = \frac{1}{2\pi i} \int_{-\pi}^{\pi} dk \partial_k \log[\omega - (t \sin k + i\Delta \cos k)] + \frac{1}{2\pi i} \int_{-\pi}^{\pi} dk \partial_k \log[\omega - (t \sin k - i\Delta \cos k)] = 0. \quad (2)$$

This expression vanishes identically due to the oppositely-signed Δ term, which ensures that one band always winds clockwise and the other counterclockwise. This seems to suggest that our system is topologically trivial. However, as we now explain, there is a well-defined and formal sense in which the model studied in our work is topologically nontrivial. This formal argument can be colloquially understood by asking and answering the question: do we define our topological invariant as the sum of the two winding numbers or as the individual winding numbers?

To answer this question, let us first point out that the two bands produced by Eq. (1) have a simple physical interpretation. For the ideal BKC model, seeing as the \hat{x} and \hat{p} quadratures are dynamically decoupled, one band corresponds to the \hat{x} degrees of freedom, and the other to the \hat{p} degrees of freedom. The fact that the sum of their winding numbers vanishes is simply a consequence of their opposite chirality with \hat{x} quadratures being amplified/deamplified when propagating to the right/left while the opposite is true of the \hat{p} quadratures. Indeed as was pointed out in Ref. [13], the BKC maps onto two uncoupled opposite-chirality Hatano-Nelson chains. This remains true even in the presence of disorder, as long as the disorder does not dynamically couple the quadratures. Still, just as we saw above, the sum of the two winding numbers proposed in Ref. [12] will vanish. Consequently, one would conclude, based on that definition, that the BKC model never possesses any nontrivial topology. However, this conclusion seems clearly pathological in this case: two uncoupled copies of a topologically nontrivial model (the Hatano-Nelson model [14]) are also topologically nontrivial.

The above discussion makes it clear that the definition in Ref. [12] does not apply if one is interested in specific symmetry-preserving perturbations, i.e. if the two Hatano-Nelson chains are not coupled. That is, if it is assumed that the disorder under consideration does not couple two distinct bands, it does not seem reasonable to define a topological invariant which ignores this fact and instead combines the topological properties of the uncoupled bands. The problem is reminiscent of the more standard Hermitian classification of topological insulators and superconductors in the presence of time-reversal symmetry (TRS) in 2 dimensions. In this context, one can explicitly show that the Chern number of a band always vanishes due to the cancellation of contributions from the different k points in the Brillouin zone connected by TRS.

As we now show, this connection to symmetry, and our claim that the BKC is topologically nontrivial, can in fact be made more formal by using the topological invariant defined in Ref. [11]. Ref. [11] predicts a \mathbb{Z}_2 NHSE in a non-Hermitian fermionic system enriched by time-reversal symmetry. There the Bloch Hamiltonian $H(k)$ satisfies $TH^T(k)T^{-1} = H(-k)$, where the time-reversal operator \hat{T} is represented by the unitary matrix T , with $TT^* = -1$ and the fermion operators transforming as $\hat{T}\hat{\psi}\hat{T}^{-1} = T\hat{\psi}$. The winding number is trivial in this case, because the contributions from $-\pi < k < 0$ and $0 < k < \pi$ always cancel. However, we can define a \mathbb{Z}_2 topological invariant $\nu(\omega) \in \{0, 1\}$ on half of the first Brillouin zone:

$$(-1)^{\nu(\omega)} = \text{sgn} \left\{ \frac{\text{Pf}[(H(\pi) - \omega)T]}{\text{Pf}[(H(0) - \omega)T]} \exp \left\{ -\frac{1}{2} \int_{k=0}^{k=\pi} dk \frac{d}{dk} \ln \det \{ [H(k) - \omega]T \} \right\} \right\}, \quad (3)$$

where for the time-reversal invariant momenta $k = 0$ and $k = \pi$, the matrix $[H(k) - \omega]T$ is skew-symmetric so that a Pfaffian can be defined.

Similar considerations also apply to our BKC model. Here, assuming a local pairing term of strength g_s , the Hamiltonian of the periodic chain reads

$$\hat{\mathcal{H}}_{\text{B,P}} = \frac{1}{2} \sum_k \begin{pmatrix} a_k \\ a_{-k}^\dagger \end{pmatrix}^\dagger \mathbf{h}_{\text{B}}(k) \begin{pmatrix} a_k \\ a_{-k}^\dagger \end{pmatrix}, \quad (4)$$

where the k sum goes over the first Brillouin zone, and

$$\mathbf{h}_{\text{B}}(k) = \begin{pmatrix} t \cos(\varphi_t - k) & i(\frac{g_s}{2} + \Delta \cos k) \\ -i(\frac{g_s}{2} + \Delta \cos k) & t \cos(\varphi_t + k) \end{pmatrix}. \quad (5)$$

Quite generally, we have the particle-hole symmetry $\sigma_x \mathbf{h}_{\text{B}}^T(k) \sigma_x = \mathbf{h}_{\text{B}}(-k)$, where σ_x is a Pauli matrix in the Nambu space. In the $g_s = 0$ model, we have an additional symmetry under the staggered local gauge transformation $\hat{U} \hat{a}_n \hat{U}^\dagger = (-1)^n \hat{a}_n$, namely

$$\hat{U} \hat{\mathcal{H}}_{\text{B,P}} \hat{U}^\dagger = -\hat{\mathcal{H}}_{\text{B,P}}. \quad (6)$$

For the Bloch Hamiltonian, this translates to $\mathbf{h}_{\text{B}}(k) = -\mathbf{h}_{\text{B}}(k + \pi)$. Making use of the particle-hole symmetry, we find the dynamical matrix

$$\tilde{\mathbf{h}}_{\text{B}}(k) = \sigma_z \mathbf{h}_{\text{B}}(k) = \begin{pmatrix} t \cos(\varphi_t - k) & i\Delta \cos k \\ i\Delta \cos k & -t \cos(\varphi_t + k) \end{pmatrix} \quad (7)$$

satisfies

$$\sigma_y \tilde{\mathbf{h}}_{\text{B}}^T(k) \sigma_y = \tilde{\mathbf{h}}_{\text{B}}(\pi - k). \quad (8)$$

The symmetry Eq. (8) ensures the possibility of a \mathbb{Z}_2 topological invariant $\nu(\omega)$. Applying Eq. (3) to our case (up to a trivial shift in momentum space), we have

$$(-1)^{\nu(\omega)} = \text{sgn} \left\{ \frac{\text{Pf}[(\tilde{\mathbf{h}}_{\text{B}}(\frac{\pi}{2}) - \omega)\sigma_y]}{\text{Pf}[(\tilde{\mathbf{h}}_{\text{B}}(-\frac{\pi}{2}) - \omega)\sigma_y]} \exp \left\{ -\frac{1}{2} \int_{k=-\frac{\pi}{2}}^{k=\frac{\pi}{2}} dk \frac{d}{dk} \ln \det\{[\tilde{\mathbf{h}}_{\text{B}}(k) - \omega]\sigma_y\} \right\} \right\}. \quad (9)$$

It is straightforward to verify that $\nu(\omega) = 1$ for ω enclosed by the spectrum of the dynamical matrix in the topological phase $t|\cos \varphi_t| < \Delta$, and $\nu(\omega) = 0$ otherwise. We note that the discussion above is not affected by uniform onsite dissipation κ apart from a translation in the complex energy plane.

Further, as the Referee suggests, robustness against disorder serves as an important indicator of nontrivial topology. Based on the above analysis, we expect our system to be topologically protected against types of disorder that preserve the symmetry Eq. (6), e.g., disorder in hopping t and nearest-neighbor pairing Δ . In fact, this was analytically demonstrated in Ref. [13] for the special case of $\phi_t = \pi/2$ (see Sec. XIII and Appendix E). In contrast, there is no protection against symmetry-breaking disorder, e.g., disorder in detuning $\delta\omega$. We note that the fact that topological models are immune to some types of disorder and not others is not uncommon. Similar to above, the paradigmatic nontrivial topology of the Hermitian SSH model is immune to disorder in the hopping strength, but not to disorder in the onsite energies within the dimer pairs. This is to be expected: the former preserves the chiral (sublattice) symmetry used to define the topological invariant, whereas the latter does not. Nevertheless, the SSH model can be topologically nontrivial in a well-defined manner.

This picture is supported by the numerical results presented in Fig. 1. We study an ensemble of disordered N -site open chains. Here, the nearest-neighbor hopping between sites j and $j+1$ is $t_{j,j+1} = t + \sigma_t \eta_{j,j+1}^t$, the nearest-neighbor pairing between sites j and $j+1$ is $\Delta_{j,j+1} = \Delta + \sigma_\Delta \eta_{j,j+1}^\Delta$, and the detuning on site j is $\delta\omega_j = \sigma_\delta \eta_j^\delta$, with $\eta_{j,j+1}^t$, $\eta_{j,j+1}^\Delta$ and η_j^δ independent standard normal random variables. Following Ref. [10] we quantify the NHSE by the inverse participation ratio (IPR) of eigenstate wave functions. Here we adopt a slightly modified definition of the IPR, $\text{IPR}^s = \sum_j (|\psi_{s,j}^p|^2 + |\psi_{s,j}^h|^2) / [\sum_j (|\psi_{s,j}^p|^2 + |\psi_{s,j}^h|^2)]^2$, where $\psi_{s,j}^{p(h)}$ is the particle (hole) wave function on site j for eigenstate s . $\text{IPR} = 1$ for a completely localized eigenstate, while $\text{IPR} = 1/N$ for a completely delocalized eigenstate. In the presence of the NHSE, we expect a broad IPR distribution; on the other hand, a sharp IPR distribution whose mean scales as $1/N$ implies the eigenstates are largely delocalized. In Fig. 1, the blue histogram shows the IPR distribution of a $\Delta = 0.3t$, $N = 100$ chain with no disorder, normalized to probability densities. The remaining two

FIG. 1. Inverse participation ratio (IPR) distributions of a 100-site open chain with different types of disorders. The blue histogram shows the IPR without disorder, the orange one corresponds to moderately strong disorder in both hopping and nearest-neighbor pairing, and the green one corresponds to weak disorder in detuning. A broad distribution indicates the NHSE and thus evidence for topological protection.

distributions describe the eigenstate IPRs of 1000 disorder realizations with different types of disorders; the broad orange one has moderately strong disorder in both hopping and nearest-neighbor pairing, while the narrow green one has a very weak disorder in the detuning. One clearly sees that the NHSE persists for hopping and pairing disorder but not detuning disorder, which is consistent with the picture that our model is topologically protected against the former but not the latter.

In the new version of the manuscript, we have added a condensed version of the above discussion to the Supplemental Information.

To the main text, we have also added endnote [43] which reads: “As discussed in detail in the Supplemental Information, we can identify a nonzero winding number in the system when considering the bands of the x and p quadratures separately, as is justified by the symmetry of the model. A \mathbb{Z}_2 topological invariant can be defined if we wish to consider the two bands together.”

In addition, in the main text, we have clarified that the “nonzero winding” refers to the band of either quadrature. To that end we made the following changes:

1. In the paragraph before Eq. (1), we modified the sentence

“Furthermore, it has a topologically nontrivial phase where the bulk spectrum of the dynamical matrix has nonzero winding in the complex energy plane.”

to read

“Furthermore, it has a topologically nontrivial phase where *each band of* the bulk spectrum of the dynamical matrix has nonzero winding in the complex energy plane.”

2. In the caption of Fig. 1(d), we modified the sentence

“For periodic boundary conditions, the spectrum has nonzero topological winding when $|\cos \varphi_t| < \Delta/t$.”

to read

“For periodic boundary conditions, *each band of the spectrum has nonzero topological winding when $|\cos \varphi_t| < \Delta/t$.*”

3. In the paragraph enclosing Eq. (6), we modified the sentence

“In the topological phase $t|\cos \varphi_t| < \Delta$, Eq. (6) forms an ellipse in the complex energy plane, yielding a nonzero winding number around the point $E = -i\kappa/2$.”

to read

“In the topological phase $t|\cos \varphi_t| < \Delta$, Eq. (6) forms an ellipse in the complex energy plane, yielding a nonzero winding number around the point $E = -i\kappa/2$ for each band.”

2) Another important point I ask the authors to clarify is the claim that the system can show non-Hermitian topology “even in the absence of dissipation”. I agree that the presence of pairing terms leads to a dynamical matrix that is non-Hermitian, but as far as I see from the theoretical description in Eqs. (4), (6) and (7), if local dissipation κ vanishes, then the resulting phase is always unstable. It is very important that the authors clarify this further since stability is an unavoidable physical requirement, and therefore would make dissipation as essential as pairing terms of the realization of the BKC model. In addition, I note that all experimental data is shown for κ on the order of the hopping t so this also does not help to make this point clearer. Finally, a small local κ will be always present in the model due to practical reasons for manipulating the system via external transmission lines (whose couplings contribute to the dissipation terms). I suggest the authors to comment on all these aspects.

We agree with the Referee’s comments that our experimental data shows κ comparable to the hopping t , and that κ will always be present in our setup due to the external transmission lines used for control and readout.

On the other hand, when the local dissipation vanishes, we would like to clarify that the BKC can be stable in the topologically nontrivial phase $t|\cos \varphi_t| < \Delta < t$. To be concrete, as long as the pairing strength is below threshold $\Delta < t$, the open boundary spectrum [Eq. (4)] is always real for $\kappa = 0$, meaning that the system is stable under open boundary conditions [13]. Furthermore, the open chain wave function in the topologically nontrivial phase, Eq. (5), is completely independent of κ and remains valid when $\kappa = 0$. Since Eq. (5) predicts the localization of wave functions which is an important aspect of the non-Hermitian skin effect, we conclude that the system does indeed show non-Hermitian topology in the absence of dissipation.

To summarize, even though our system is unavoidably subject to dissipation in practice, we maintain that dissipation is not a necessary ingredient in realizing effective non-Hermiticity and non-Hermitian topology in quadratic bosonic Hamiltonians.

3) My third concern has to do with the statements “genuinely quantum simulation” or “genuinely quantum conditions” found in several places in the text. I wonder to what extent the presented simulation of the model is quantum. As far I understand the authors send and measure coherent (Gaussian fields) which can be equally well described by a classical model of linearly coupled modes such as in Naaman & Aumentado, PRX Quantum 3, 020201 (2022). I think the key is to measure quantum fluctuations in the system, of the generated squeezed states, for instance. I think this may be possible in the discussed setup, but I do not see clearly to what extent this is reached in the present work. I, therefore, ask the authors to be clearer in this respect and to soften their claims regarding the quantum regime if this is needed.

First, we would like to clarify that by “genuinely quantum” we are referring to our approach to effective non-Hermiticity. The vast majority of experimental demonstrations of non-Hermitian systems rely on strong dissipation, which generally means that their dynamics are fully classical on relevant time scales. In stark contrast, our approach

FIG. 2. Quantum discord in the steady state of the 3-site open chain calculated using fit parameters in Tables II and III. The quantum discord characterizes the amount of nonclassical correlation in the steady state with no coherent input tones. (Left) Quantum discord (solid) and the classical part of the mutual information (dashed) for each pair of modes. (Right) Quantum discord per photon for each pair of modes.

uses the coherent processes of squeezing and antisqueezing, and ensures that the equations of motion are always unconditionally non-Hermitian at the level of quantum mechanical operators.

Second, as the Referee points out, we have focused on measuring Gaussian states in this work. Nevertheless, with downconversion we coherently generate two-mode squeezed states, which are well-known to have nonclassical correlations including entanglement. In fact, we have previously proven entanglement in similar experiments in the same parametric cavity system, specifically in Ref. [15]. In that experiment, the main intention, and key result, was to show that we could generate genuine multimode entanglement by combining multiple parametric pumps. That was not the goal of this experiment, though we have every reason to believe that the system supports nonclassical correlations.

In fact, we have experimentally measured correlations between the pairs of modes in our BKC system. Although we did not perform the necessary calibration to quantify the degree to which those correlations are quantum, we can explore that question theoretically based on our model and the experimentally extracted fit parameters. Using the model, we numerically compute the steady state covariance matrix of the 3-site system in the absence of coherent input tones [16], and calculate the quantum discord and the mutual information of each pair of cavity modes as a two-mode Gaussian state [17]. The quantum discord is roughly defined as the full mutual information between the modes minus its classical part and is a measure of genuinely quantum correlations. From the results plotted in Fig. 2, one can see that the amounts of classical and purely quantum correlations are comparable in the steady state. Since discord is normalized to “units” of quantum information, it is meaningful to look at the discord (units of quantum information) per photon. Based on our model calculations, there are $0.1 \sim 0.3$ units of genuinely quantum information per photon for the two-mode states ab and bc , which is not insignificant.

In terms of revision to the manuscript, in the Discussion section, we modified the sentence:

“Furthermore, since the dynamics are the results of a coherent process in a Hermitian Hamiltonian, our AQS platform can implement genuine quantum dynamics with effective non-Hermiticity.”

to read:

“Furthermore, since the dynamics are the results of a coherent process, *as opposed to dissipation*, in a Hermitian Hamiltonian, our AQS platform can implement genuine quantum dynamics with effective non-Hermiticity.”

4) Another important comment is regarding the various approaches to non-Hermitian topology highlighted in the introduction. Only two options are mentioned: either take an open quantum system and post-select the dynamics condition to the absence of jumps or consider a Hermitian system with squeezing terms as described in the present work. Here, the authors are missing a relevant approach, which is considering a general open quantum system (with not only pairing terms but also non-local pumping or dissipation) and analyzing its non-trivial topology by mapping to topological insulator theory via the singular value decomposition and the doubled Hamiltonian [Gong et al. PRX 8, 031079 (2018), Porras et al. PRL 122, 143901, (2019), Herviou, et al. Phys. Rev. A 99, 052118 (2019), Okuma et al. Phys. Rev. B 102, 014203 (2020)]. This approach includes the non-Hermiticity via pairing terms as a special case, and it is more general as it allows the classification of non-trivial topological phases according to the underlying symmetries.

We thank the Referee for pointing to these references.

The Referee brings up a valid point: the dynamics of a general open quantum system can be thought of as being generated by some non-Hermitian Hamiltonian, i.e. the Lindbladian. Consequently, the averages of specific observables of interest, such as particle number or steady-state current, can also be thought of as generated by some non-Hermitian Hamiltonian. Nevertheless, the doubled-Hamiltonian approach fundamentally relies on the noninteracting band theory. When one considers noninteracting bosonic models without any pairing terms, these equations of motion and the corresponding classification schemes are, in a well-defined manner, *classical* in nature. Indeed after specifying the quadratic Hamiltonian and linear bosonic dissipators, one can always come up with an equivalent classical model of coupled harmonic oscillators subject to noisy forces such that the equations of motion for the classical correlation function exactly mimic the quantum equations of motion [18]. Dissipative non-interacting fermions can also be interpreted in an analogous manner; Pauli exclusion is enforced via constraints on the anti-Hermitian part of the Hamiltonian and the correlation matrix characterising the noise [18].

Therefore, although these schemes are interesting and useful in their own right, given that our work focuses on the quantum nature of the model at hand (e.g. the possibility of creating squeezed states), we have decided to focus on the other two approaches. We have, however, added a sentence in the introduction indicating that these two approaches are not an exhaustive list of ways to generate non-Hermitian dynamics.

In the revised manuscript, we have added the following sentence to the Introduction section: “There are various approaches to non-Hermitian dynamics [14, 19–21].”

5) Finally, when commenting on applications of the system to topological amplification, the list does not include relevant references in the field such as Wanjura et al., Nat. Commun. 11, 3149 (2020) and Ramos et al. PRA 103, 033513 (2021).

We thank the Referee for pointing to these references. They are now cited in the Discussion section in the main text.

In the revised manuscript, in the Discussion section, we added Refs. [22, 23] to the sentence:

“We can utilize the chiral features as a phase-dependent quantum amplifier [13].”

Other changes

In addition to the changes in response to the Referees' comments, we have also made the following changes in the revised manuscript.

1. Immediately after Eq. (6) in the main text, the sentence

“where $k_n = n\pi/N$ if N is odd and $k_n = 2n\pi/N$ if N is even, $n = 1, 2, \dots, 2N$.”

now reads

“where $k_n = 2n\pi/N$, $n = 1, 2, \dots, N$.”

This change does not affect the rest of the manuscript.

2. In the paragraph before Eq. (1), we added Ref. [24] to the sentence:

“This is accompanied by the remarkable property of the NHSE [11, 25, 26]: ...”

3. In the Introduction section, we added Ref. [18] to the sentence:

“... conditioned on the absence of a quantum jump, state evolution is described by an effective non-Hermitian Hamiltonian [27].”

-
- [1] K. Wang, A. Dutt, K. Y. Yang, C. Wojcik, J. Vučković, and S. Fan, *Science* **371**, 1240 (2021).
 - [2] K. Wang, A. Dutt, C. C. Wojcik, and F. S., *Nature* **598**, 59–64 (2021).
 - [3] Y.-P. Wang, W. Wang, Z.-Y. Xue, W.-L. Yang, Y. Hu, and Y. Wu, *Scientific reports* **5**, 8352 (2015).
 - [4] P. Roushan, C. Neill, A. Megrant, Y. Chen, R. Babbush, R. Barends, B. Campbell, Z. Chen, B. Chiaro, A. Dunsworth, et al., *Nature Physics* **13**, 146 (2017).
 - [5] S. De Léséleuc, V. Lienhard, P. Scholl, D. Barredo, S. Weber, N. Lang, H. P. Büchler, T. Lahaye, and A. Browaeys, *Science* **365**, 775 (2019).
 - [6] K. Kawabata, K. Shiozaki, and S. Ryu, *Physical Review B* **105**, 165137 (2022).
 - [7] S.-B. Zhang, M. M. Denner, T. c. v. Bzdušek, M. A. Sentef, and T. Neupert, *Physical Review B* **106**, L121102 (2022).
 - [8] W. N. Faugno and T. Ozawa, *Physical Review Letters* **129**, 180401 (2022).
 - [9] N. Okuma and M. Sato, *Annual Review of Condensed Matter Physics* **14**, 83 (2023).
 - [10] G.-F. Guo, X.-X. Bao, H.-J. Zhu, X.-M. Zhao, L. Zhuang, L. Tan, and W.-M. Liu, *Communications Physics* **6**, 363 (2023).
 - [11] N. Okuma, K. Kawabata, K. Shiozaki, and M. Sato, *Physical Review Letters* **124**, 086801 (2020).
 - [12] Á. Gómez-León, T. Ramos, A. González-Tudela, and D. Porras, *Quantum* **7**, 1016 (2023).
 - [13] A. McDonald, T. Pereg-Barnea, and A. Clerk, *Physical Review X* **8**, 041031 (2018).
 - [14] Z. Gong, Y. Ashida, K. Kawabata, K. Takasan, S. Higashikawa, and M. Ueda, *Physical Review X* **8**, 031079 (2018).
 - [15] C. S. Chang, M. Simoen, J. Aumentado, C. Sabín, P. Forn-Díaz, A. Vadiraj, F. Quijandría, G. Johansson, I. Fuentes, and C. Wilson, *Physical Review Applied* **10**, 044019 (2018).
 - [16] T. Prosen and T. H. Seligman, *Journal of Physics A: Mathematical and Theoretical* **43**, 392004 (2010).
 - [17] G. Adesso and A. Datta, *Physical Review Letters* **105**, 030501 (2010).
 - [18] A. McDonald, R. Hanai, and A. A. Clerk, *Phys. Rev. B* **105**, 064302 (2022).
 - [19] D. Porras and S. Fernández-Lorenzo, *Phys. Rev. Lett.* **122**, 143901 (2019).
 - [20] L. Herviou, J. H. Bardarson, and N. Regnault, *Phys. Rev. A* **99**, 052118 (2019).
 - [21] N. Okuma and M. Sato, *Phys. Rev. B* **102**, 014203 (2020).
 - [22] C. C. Wanjura, M. Brunelli, and A. Nunnenkamp, *Nature communications* **11**, 3149 (2020).
 - [23] T. Ramos, J. J. García-Ripoll, and D. Porras, *Physical Review A* **103**, 033513 (2021).
 - [24] N. Okuma and M. Sato, *Phys. Rev. B* **103**, 085428 (2021).
 - [25] S. Yao and Z. Wang, *Physical Review Letters* **121**, 086803 (2018).
 - [26] G. Lee, A. McDonald, and A. Clerk, *Physical Review B* **108**, 064311 (2023).
 - [27] A. J. Daley, *Advances in Physics* **63**, 77 (2014).

REVIEWERS' COMMENTS

Reviewer #3 (Remarks to the Author):

I thank very much the authors for the very careful and detailed response. I have read all explanations to my questions and concerns and I think that they are all satisfactory. The addition to the supplementary material regarding the detailed analysis of the topology of the model is very clear as well. I would only suggest to make some connection or reference in the main text to the discussion of the disorder found in the supplementary, as otherwise these may be hard to find for a reader.

In general, I think the paper is of high quality and interesting for the field of quantum simulation of topological models and I recommend publication in Nature Communications.

As a side comment to the authors, I wanted to stress that the method of the extended Hamiltonian and singular value decomposition is not necessary classical as they argued, but it is a general method that can be applied to any non-interacting quantum system as well, including local and non-local pairing terms.